# Endothelial lipase mediates efficient lipolysis of triglyceride-rich lipoproteins

**Sumeet A. Khetarpal**[1,2☯], **Cecilia Vitali**[1☯], **Michael G. Levin**[3,4,5], **Derek Klarin**[6,7,8], **Joseph Park**[1], **Akhil Pampana**[2,7,8,9], **John S. Millar**[4], **Takashi Kuwano**[1], **Dhavamani Sugasini**[10], **Papasani V. Subbaiah**[10], **Jeffrey T. Billheimer**[4], **Pradeep Natarajan**[2,7,8,9], **Daniel J. Rader**[1]*

1 Departments of Medicine and Genetics, Perelman School of Medicine, University of Pennsylvania, Philadelphia, Pennsylvania, United States of America, 2 Cardiovascular Research Center, Massachusetts General Hospital, Boston, Massachusetts, United States of America, 3 Division of Cardiovascular Medicine, Perelman School of Medicine, University of Pennsylvania, Philadelphia, Pennsylvania, United States of America, 4 Department of Medicine, Perelman School of Medicine, University of Pennsylvania, Philadelphia, Pennsylvania, United States of America, 5 Corporal Michael J. Crescenz VA Medical Center, Philadelphia, Pennsylvania, United States of America, 6 Boston VA Healthcare System, Boston, Massachusetts, United States of America, 7 Center for Genomic Medicine, Massachusetts General Hospital, Harvard Medical School, Boston, Massachusetts, United States of America, 8 Program in Medical and Population Genetics, Broad Institute of MIT and Harvard, Cambridge, Massachusetts, United States of America, 9 Department of Medicine, Harvard Medical School, Boston, Massachusetts, United States of America, 10 Section of Endocrinology, Department of Medicine, University of Illinois at Chicago; Jesse Brown VA Medical Center, Chicago, Illinois, United States of America

☯ These authors contributed equally to this work.
* rader@pennmedicine.upenn.edu

**Data Availability Statement:** Genetic association data from the UK Biobank (UKBB), Million Veteran Program (MVP) and Global Lipids Genetics Consortium (GLGC) are reported in the manuscript. Raw data corresponding to analyses from the UKBB can be accessed upon application

## Abstract

Triglyceride-rich lipoproteins (TRLs) are circulating reservoirs of fatty acids used as vital energy sources for peripheral tissues. Lipoprotein lipase (LPL) is a predominant enzyme mediating triglyceride (TG) lipolysis and TRL clearance to provide fatty acids to tissues in animals. Physiological and human genetic evidence support a primary role for LPL in hydrolyzing TRL TGs. We hypothesized that endothelial lipase (EL), another extracellular lipase that primarily hydrolyzes lipoprotein phospholipids may also contribute to TRL metabolism. To explore this, we studied the impact of genetic EL loss-of-function on TRL metabolism in humans and mice. Humans carrying a loss-of-function missense variant in *LIPG*, p.Asn396-Ser (rs77960347), demonstrated elevated plasma TGs and elevated phospholipids in TRLs, among other lipoprotein classes. Mice with germline EL deficiency challenged with excess dietary TG through refeeding or a high-fat diet exhibited elevated TGs, delayed dietary TRL clearance, and impaired TRL TG lipolysis *in vivo* that was rescued by EL reconstitution in the liver. Lipidomic analyses of postprandial plasma from high-fat fed *Lipg⁻/⁻* mice demonstrated accumulation of phospholipids and TGs harboring long-chain polyunsaturated fatty acids (PUFAs), known substrates for EL lipolysis. *In vitro* and *in vivo*, EL and LPL together promoted greater TG lipolysis than either extracellular lipase alone. Our data positions EL as a key collaborator of LPL to mediate efficient lipolysis of TRLs in humans and mice.

to the UK Biobank (https://www.ukbiobank.ac.uk/using-the-resource/). Raw association data from MVP are available through dbGaP (the database of Genotypes and Phenotypes; accession code: phs001672.v2.p1). Raw data corresponding to analyses from the GLGC are available from http://csg.sph.umich.edu/willer/public/lipids2017/. All other raw data presented in the manuscript is provided in the Supporting Information files associated with the manuscript.

**Funding:** This work was supported by National Institutes of Health grants HL055323 (DJR) and F30HL124967 (SAK), and American Heart Association grant 18POST34080184 (CV). The funders had no role in study design, data collection and analysis, decision to publish, or preparation of the manuscript.

**Competing interests:** I have read the journal's policy and the authors of this manuscript have the following competing interests: Co-founder, Staten Biotechnology (DJR)

## Author summary

Endothelial lipase (EL) plays a pivotal role in the breakdown of high-density lipoproteins (HDLs) by hydrolyzing phospholipids on HDL surfaces. Here we show through studies of humans and mice with genetic loss-of-function of EL activity that EL is also crucial in catabolizing triglyceride (TG)-rich lipoproteins, particularly in states of nutrient excess such as after refeeding or after feeding a high-fat diet. We demonstrate that EL collaborates with lipoprotein lipase (LPL), the predominant enzyme hydrolysing triglycerides, to promote the breakdown of triglycerides on lipoproteins. The mechanism for this may be due to an underappreciated ability of EL to hydrolyze TGs and cooperate with LPL to efficiently break down and clear these particles. Our data adds important insights into the mechanisms of circulating TG metabolism in mice and humans as informed by large-scale human genetics.

## Introduction

Lipolysis of circulating triglyceride (TG)-rich lipoproteins (TRLs) is critical for the delivery of fatty acids to tissues and control of circulating TG levels [1, 2]. TRLs are also proatherogenic lipoproteins that directly contribute to cardiovascular risk [3, 4]. Recent insights from epidemiology and human genetics have shown that high TRL levels increase cardiovascular risk while lower TG levels are associated with risk reduction independently of low-density lipoprotein cholesterol (LDL-C) levels [4, 5]. Human genetic studies in particular have supported the notion that augmenting TRL lipolysis protects from vascular risk, leading to ongoing development of multiple pharmaceutical approaches targeting TRLs [5].

Lipoprotein lipase (LPL) is the primary effector of TG lipolysis on TRLs [1, 6]. LPL is highly expressed in tissues crucial for energy utilization and storage such as the heart, skeletal muscle, and adipose depots, and is translocated from sites of synthesis to the luminal surface of surrounding capillary endothelial cells where it hydrolyzes primarily TGs on circulating lipoproteins [1]. Genetic loss- or gain-of-function of *LPL* in animal models and in humans has demonstrated the key role it plays in TG clearance, with genetic loss-of-function in LPL or key adaptors leading to severe hypertriglyceridemia [1, 6, 7]. In addition to LPL, hepatic lipase (HL) is another extracellular lipase in the LPL family with a demonstrated role in TRL catabolism. Expressed in liver, HL hydrolyzes TGs and phospholipids (PLs) on TRL remnants, intermediate density lipoproteins (IDLs) and high density lipoproteins (HDLs) [8]. Inhibition or deficiency of HL in animal models results in delayed chylomicron remnant and IDL clearance *in vivo* [8] and humans with genetic loss-of-function in *LIPC*, which encodes HL, also demonstrate elevations in TRL remnants [9].

Endothelial lipase (EL) is an endothelially-derived member of the LPL gene family based on its primary sequence [10, 11] and is structurally similar to LPL [12, 13]. In contrast to LPL, EL has primarily phospholipase activity and has a high affinity for HDL [14, 15]. Overexpression of EL in mice reduces HDL-C and promotes HDL catabolism [11, 16] and antibody inhibition or genetic deletion in mice raises HDL-C levels [17, 18]. This work in mice is further supported by human genetics, with carriers of loss-of-function variants in the *LIPG* gene encoding EL demonstrating markedly elevated HDL-C levels and other measures of decreased HDL catabolism [19, 20]. However, EL has also been shown to have phospholipase activity on low density lipoproteins (LDL) in vitro [14, 21] and to promote LDL catabolism in vivo [22]. Like LPL, EL is inhibited by ANGPTL3 and recent studies suggest that increased EL activity contributes to the reduction of LDL-C levels in the setting of ANGPTL3 inhibition or silencing

[23, 24]. However, a direct role of EL in the metabolism of apoB-containing triglyceride-rich lipoproteins has not been elucidated.

Importantly, a functional relationship between LPL and EL *in vivo* was suggested by the observation that adipose-specific LPL deletion in mice caused a profound upregulation in EL expression and phospholipase activity in these tissues, thereby augmenting TRL-mediated fatty acid uptake in the setting of LPL deficiency [25]. Given that EL is expressed throughout the vascular endothelium where LPL is located [26], we hypothesized that EL may collaborate with LPL and contribute to the lipolysis of TRLs *in vivo*. Here, we studied human and murine genetic EL loss-of-function to test the hypothesis that EL is required for efficient metabolism of TRLs.

## Results

### Genetically reduced function of LIPG is associated with elevated TG and TG-rich lipoproteins in humans

Previous resequencing studies of *LIPG* in humans with elevated HDL-C have demonstrated that the low-frequency p.Asn396Ser missense variant (rs77960347) is a loss-of-function variant that blunts HDL catabolism by EL through impairing its lipolytic activity and is robustly associated with elevated HDL-C in humans [19, 20]. This variant has a minor allele frequency of approximately 1.0–1.2% in European individuals (gnomAD). In a previous meta-analysis of four European cohorts comprising 25,012 individuals, this variant was associated with elevated HDL-C but was not significantly associated with plasma TG or measures of apoB-containing lipoproteins [27]. We hypothesized that more recent and larger genome-wide genotyping and sequencing efforts of this loss-of-function variant may inform on a potential relationship between *LIPG* and TG-rich lipoproteins. Mining the results of exome sequencing of ~200,000 participants in the UK BioBank (UKBB) [28], exome-wide genotyping in ~370,000 participants of the UKBB, genotyping in ~300,000 individuals by the Global Lipids Genetics Consortium (GLGC) [29], and genotyping of >200,000 individuals in the Million Veteran Program (MVP) [30], we tested the association of the *LIPG* p.Asn396Ser variant with plasma lipids and performed a fixed-effects meta-analysis of this variant across the cohorts (**Fig 1**). As previously seen in smaller cohorts, *LIPG* p.Asn396Ser was significantly associated with elevated HDL-C across the cohorts. Notably, the variant was also associated with elevated measures of apoB-containing lipoproteins rich in triglycerides, namely increased elevated TG (P = 1.36x10$^{-10}$) and LDL-C (P = 2.27x10$^{-65}$) (**Fig 1C and 1D**), consistent with a role of EL in the metabolism of apoB-containing lipoproteins.

To further explore the molecular mechanisms by which *LIPG* loss-of-function may lead to elevated TGs, we examined the relationship of the *LIPG* p.Asn396Ser variant with lipoprotein subclasses in humans. We mined a cohort of ~25,000 European individuals for whom serum lipid and lipoprotein metabolites were measured by NMR and who were genotyped for the *LIPG* p.Asn396Ser variant as part of genome-wide genotyping [31]. This analysis confirmed elevated concentrations of VLDL and IDL subclasses of TRLs in LIPG p.Asn396Ser carriers relative to noncarriers (**Fig 2A**). We explored the relationship between LIPG p.Asn396Ser and serum phospholipid concentrations in this cohort and found that this loss-of-function variant was associated with both elevated total serum phospholipids and elevated phospholipids in TRLs and HDL subclasses (**Fig 2B**). These data suggest that humans with genetic loss-of-function in *LIPG* harbor elevated TGs and that EL may modulate phospholipid metabolism in TRLs in addition to its known role in HDL catabolism.

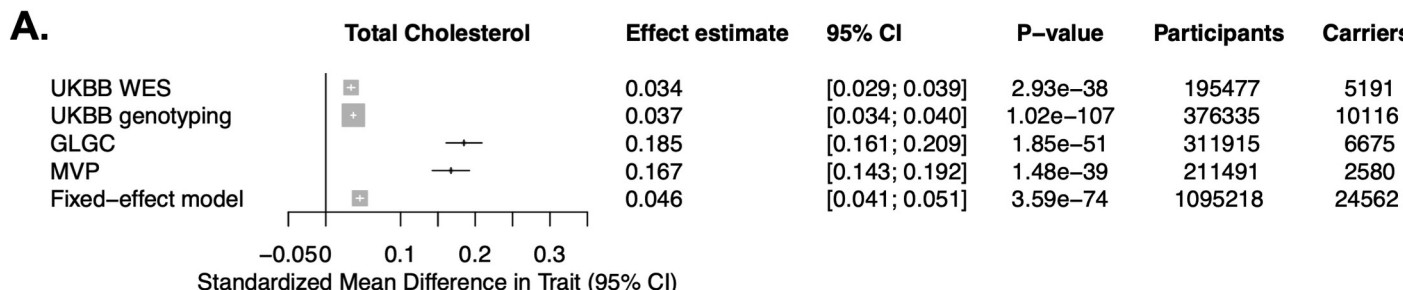

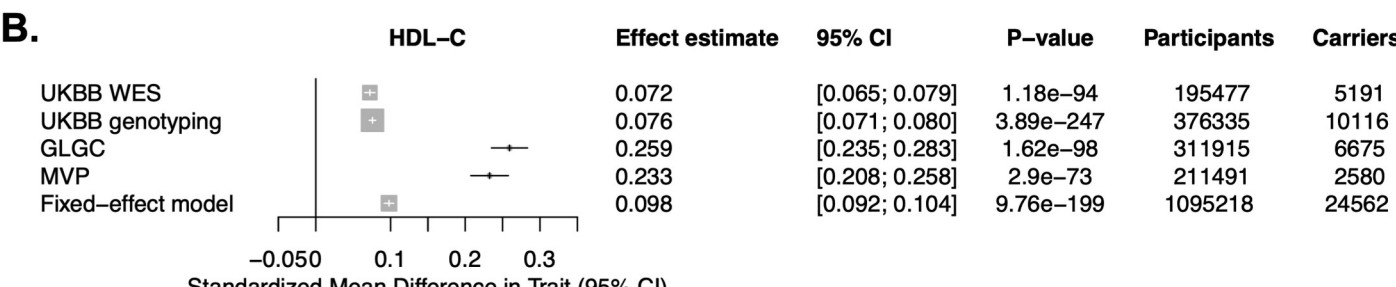

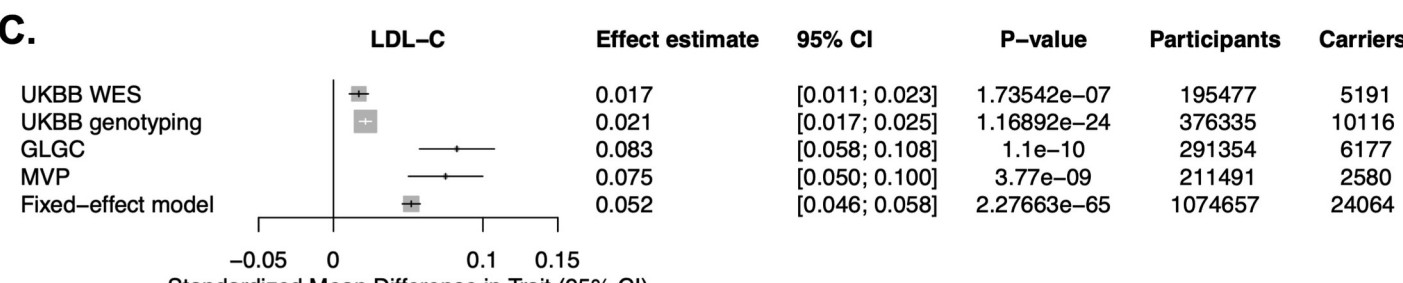

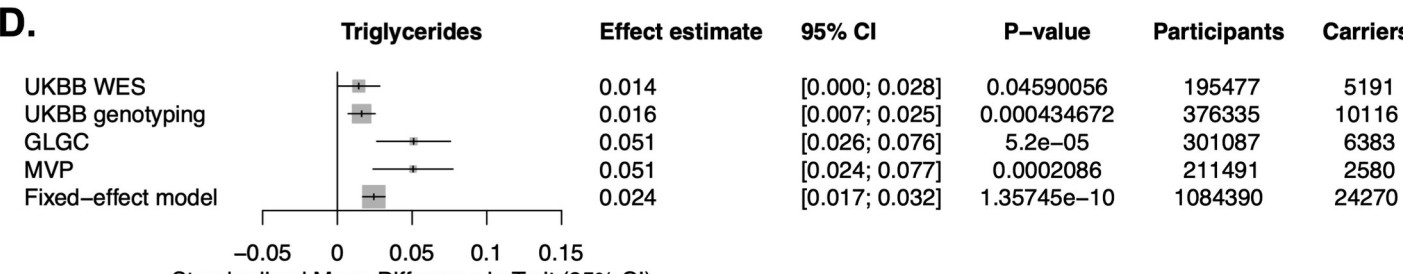

**Fig 1. Association of the *LIPG* p.Asn396Ser variant with plasma lipids in humans. A-D**. Association of *LIPG* p.Asn396Ser with plasma total cholesterol (A), HDL-C (B), LDL-C (C), and triglycerides (D) in the UK Biobank (UKBB) whole exome sequencing subset, UKBB genome-wide genotyping subset, Global Lipids Genetics Consortium (GLGC), and Million Veteran Program (MVP) cohorts, and a fixed-effects meta-analysis of these cohorts. Effect estimates after log-transformation for each lipid trait in standard deviation (S.D.) units and 95% confidence intervals are plotted.

### Deletion of EL in mice delays clearance of TG-rich lipoproteins

To further investigate the molecular mechanism by which EL regulates TRL metabolism, we studied the impact of EL deficiency in TRL metabolism *in vivo* using germline *Lipg* homozygous-deficient mice (*Lipg*$^{-/-}$ mice) [17]. We measured plasma lipids in WT vs *Lipg*$^{-/-}$ mice after fasting for four hours compared to refeeding for four hours (**Fig 3A**). In the fasting state, *Lipg*$^{-/-}$ mice have elevated total cholesterol, HDL-C, nonHDL-C, and phospholipids compared

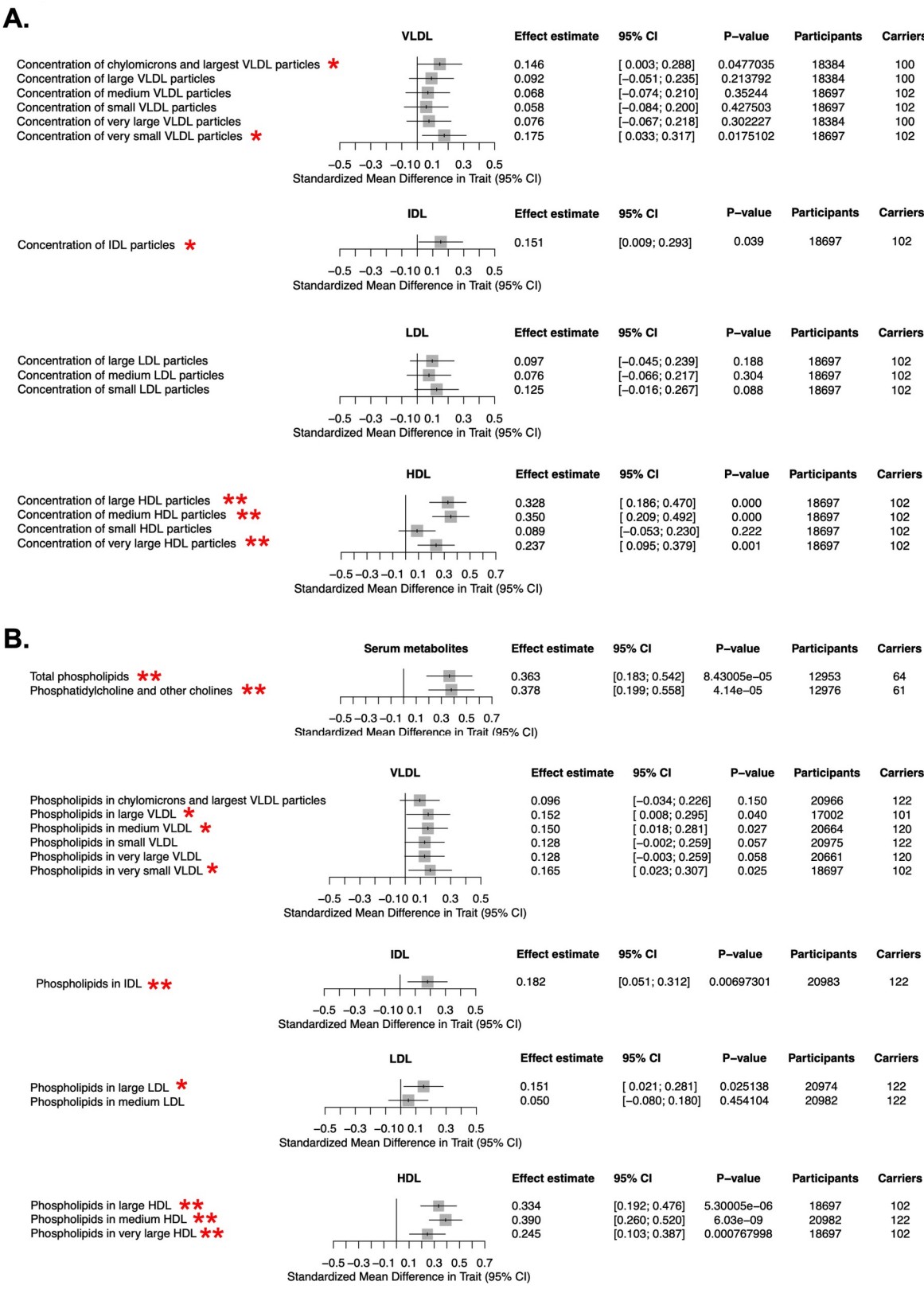

**Fig 2. Association of *LIPG* p.Asn396Ser variant with serum lipoprotein and lipid metabolites by NMR. A.** Association of *LIPG* p. Asn396Ser variant with lipoprotein subclass concentrations measured through NMR spectroscopy from human serum from a European

cohort comprising up to 24,925 individuals as described previously [31] (see Materials and Methods). **B.** association of *LIPG* p.Asn396Ser variant with phospholipid concentrations in serum and in lipoprotein subclasses from analysis in (A). Effect estimates for each log-transformed lipid trait in standard deviation (S.D.) units and 95% confidence intervals are plotted. Single red star (*) indicates measures with nominally significant associations between *LIPG* p.Asn396Ser carriers and noncarrier controls (P<0.05); double red star (**) indicates measures with experiment-wide significant differences between carriers and controls (FDR < 0.05).

to WT mice. Notably, after refeeding, *Lipg*$^{-/-}$ mice demonstrated a more than two-fold elevation in TGs relative to WT mice (**Fig 3A and 3B**). This TG increase was exclusively in the TRL fractions after FPLC fractionation (**Fig 3C**). To test whether increased VLDL-TG secretion was a possible cause of the elevated TGs in the *Lipg*$^{-/-}$ mice, we administered poloxamer P407, a competitive inhibitor of LPL, to the mice and measured plasma TG accumulation, finding that *Lipg*$^{-/-}$ mice did not have increased VLDL-TG secretion (**S1 Fig**).

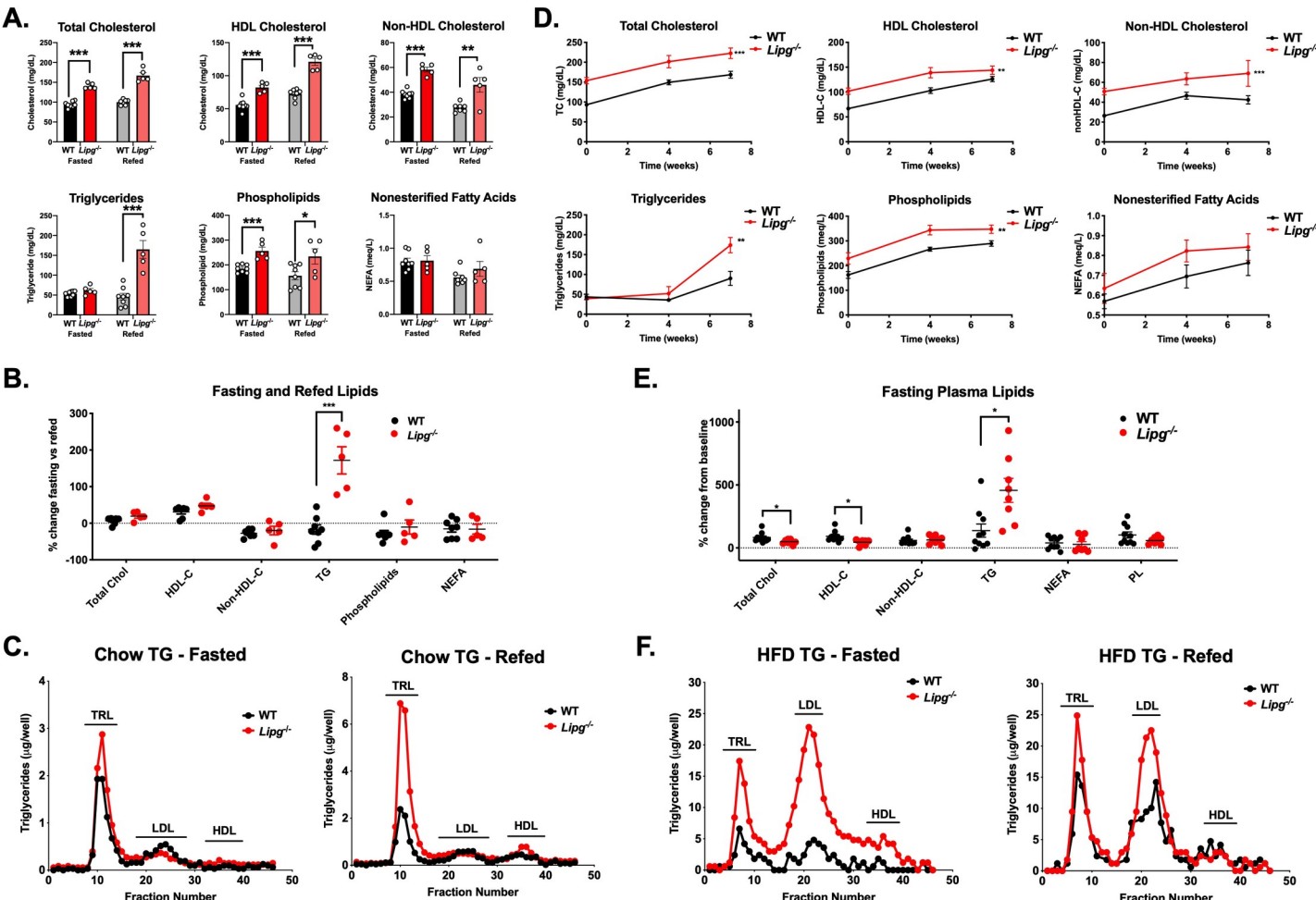

**Fig 3. Genetic loss-of-function of *Lipg* and plasma triglycerides in mice. A.** Plasma TC, HDL-C, nonHDL-C, TGs, phospholipids, and nonesterified fatty acids from WT vs. *Lipg*$^{-/-}$ mice after a 4 hour fast and again after 4 hours of refeeding a chow diet. Lipids were measured by chemical autoanalyzer. **B.** Percent change in each plasma lipid measure from (A) in 4 hour refed vs fasting measures. **C.** TG from plasma of fasted mice (left) and four hour-refed mice (right) fractionated by fast-protein liquid chromatography (FPLC) from pools from WT or *Lipg*$^{-/-}$ mice from groups in (A). **D.** Total cholesterol, HDL-C, nonHDL-C, TGs, phospholipids, and nonesterified fatty acids in WT and *Lipg*$^{-/-}$ mice after initiation of feeding adult mice a diet composed of 45% kilocalories fat for the indicated timepoints. Plasma was collected 4 hours after fasting and lipids were measured by autoanalyzer. **E.** Fasting plasma lipids after 7 weeks of high fat feeding were normalized to the levels at zero weeks of feeding and expressed as a percentage change. **F.** Triglycerides from fasted (left) and refed (mice) from (D-E) after FPLC-fractionation of plasma. A, B, E: *P<0.05, **P<0.01, ***P<0.001, Student's unpaired T-test. D: **P<0.01, ***P<0.001, repeated factor 2-way ANOVA compared to WT group. Data is expressed as mean ± S.E.M.

## A high fat diet accentuates the effects of EL deficiency on TRL catabolism

We next assessed the contribution of EL to dietary TG metabolism by feeding *Lipg*[-/-] mice a high fat diet (HFD; 45% kilocalories from saturated fat) that is known to promote hypertriglyceridemia. By week 7 of the HFD, plasma TGs were substantially increased in *Lipg*[-/-] mice compared with WT mice (**Fig 3D and 3E**). After 24 weeks of HFD feeding, *Lipg*[-/-] mice had markedly elevated TRL TGs relative to HFD-fed WT mice, particularly after refeeding (**Fig 3F**). These findings support the data from human LIPG p.Asn396Ser variant carriers suggesting a delayed clearance of remnant TRLs in the absence of EL function.

## Delayed TRL remnant clearance in Lipg[-/-] mice

To further explore the role of EL in mediating postprandial TG metabolism, we subjected *Lipg*[-/-] mice fed either chow diet or HFD to an oral fat tolerance test by administering an olive oil gavage after overnight fast and measuring plasma TGs at several time points to chart the clearance of dietary TRLs. Chow-fed *Lipg*[-/-] mice exhibited a substantial delay in the clearance of TRLs by oral fat tolerance testing compared to WT (**Fig 4A**), which was apparent even seven hours after gavage by FPLC-fractionation of plasma and measurement of the TRL TGs (**Fig 4B**). Similarly, HFD-fed *Lipg*[-/-] mice demonstrated a substantial delay in clearance of postprandial TGs with no demonstrable decrease in TGs and visible chylomicronemia for seven hours after gavage (**Fig 4C and 4D**).

To test if hepatic EL was sufficient to restore the normal clearance of circulating TRLs, we reconstituted EL expression in the livers of *Lipg*[-/-] mice using AAV vectors expressing murine *Lipg* and measured the impact on TRL clearance *in vivo*. Expression of EL in *Lipg*[-/-] mice reduced TGs and other plasma lipids to levels of WT mice over five weeks after AAV expression (**S2 Fig**). Hepatic *Lipg* reconstitution in *Lipg*[-/-] mice fed a high fat diet for 12 weeks trended to increase postprandial TG clearance as measured by OFTT compared to *Lipg*[-/-] mice treated will Null AAV (P = 0.06, 2 way ANOVA compared with WT AAV Null group, **Fig 4E**). These data collectively support the notion that EL activity promotes the clearance of TRLs, particularly post-prandially and in the setting of higher circulating plasma TGs such as diet-induced hypertriglyceridemia.

Given the role EL plays as a lipase on lipoproteins, we hypothesized the delay in remnant TRL clearance due to EL deficiency was due to a direct loss of lipolytic activity *in vivo*. We tested this hypothesis by administering $^3$H-triolein and $^{125}$I-apolipoprotein B (apoB) protein-labeled human TRLs intravenously in HFD-fed *Lipg*[-/-] and WT mice and measuring clearance of the radioisotopes in plasma over 15 minutes (**Fig 5**). *Lipg*[-/-] mice had significantly slower clearance of TRL $^3$H-triolein (**Fig 5A**) but no difference in clearance of $^{125}$I-labeled TRL apoB over this time frame (**Fig 5B**), establishing that the primary defect is TG hydrolysis. Moreover, reconstitution of hepatic EL via AAV expression of murine *Lipg* in *Lipg*[-/-] mice rescued the lipolysis of $^3$H-triolein labeled TRLs (**Fig 5C**). The decreased TG hydrolysis of TRLs was associated with significantly decreased uptake of $^3$H-oleate in the livers of *Lipg*[-/-] mice (**Fig 5D**). There was a statistically significant but smaller decrease in the uptake of $^3$H FA by brown adipose tissue, and an increase in $^3$H uptake in spleen, likely reflective of increased splenic macrophage uptake of labeled TRL and TRL remnants. Notably, the reduced hepatic uptake of TRL-derived FAs occurred despite marked compensatory upregulation of expression of hepatic LPL, but not HL (**Fig 5E and 5F**). Collectively these data support the notion that EL is an important physiological mediator of efficient TRL TG lipolysis in the HFD-fed state.

## EL and LPL collaborate to promote efficient TRL lipolysis

EL possesses a substrate preference for lipolysis of phospholipids bearing long-chain polyunsaturated FAs in the sn-2 position of HDL-phosphatidylcholine (PC) [32–34]. To further

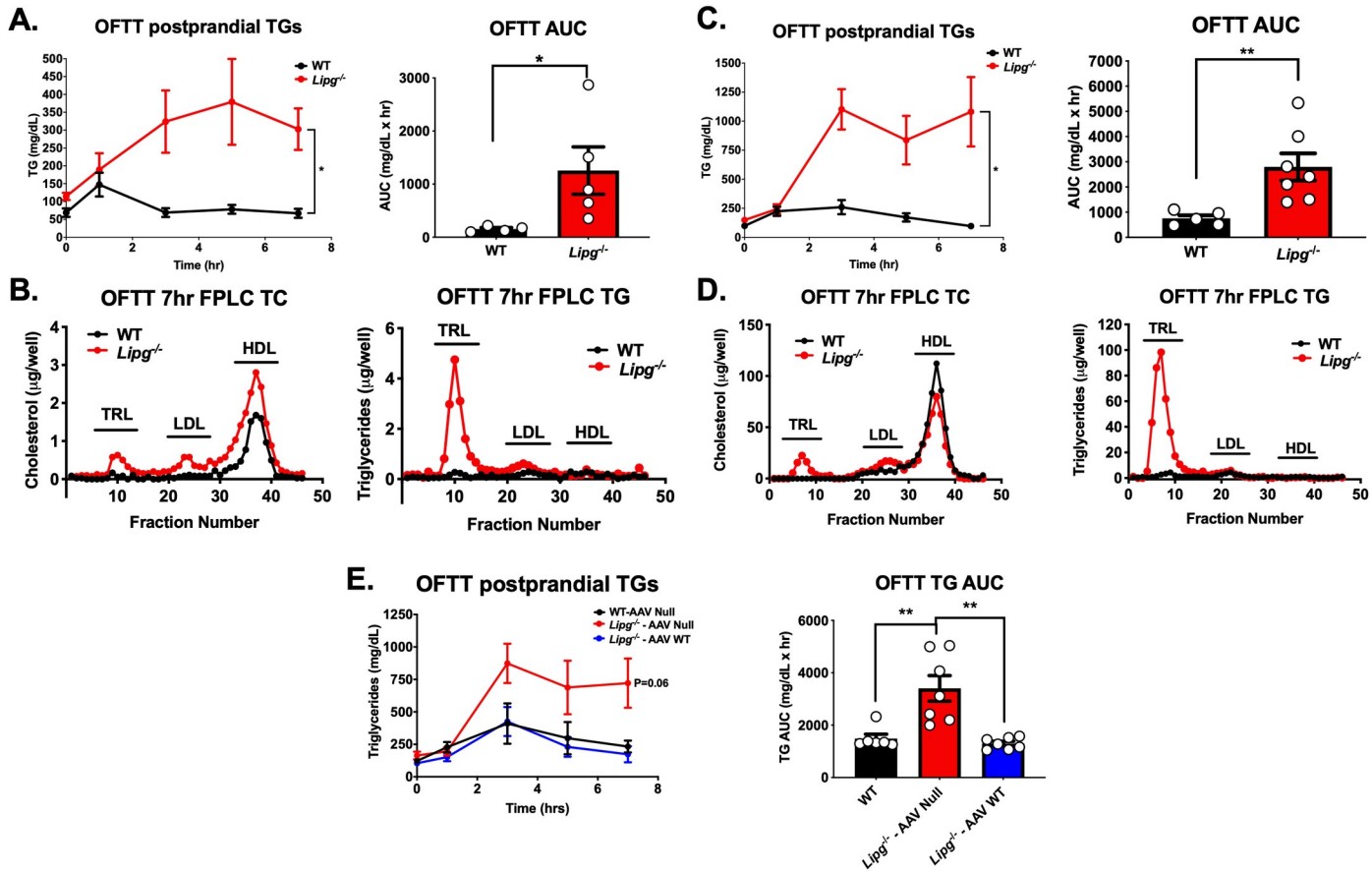

**Fig 4. Postprandial chylomicron clearance in *Lipg*-deficient mice. A.** Plasma triglycerides (left) and area under the triglyceride clearance curves (right) in EL WT vs. Lipg$^{-/-}$ mice fed a regular chow diet after overnight fasting and oral fat tolerance testing with olive oil gavage. **B.** Cholesterol and triglycerides from FPLC fractions of pooled plasma from the 7 hour timepoint from the olive oil gavage in A. **C.** Plasma triglycerides (left) and area under the triglyceride clearance curves (right) in EL WT vs. *Lipg*$^{-/-}$ mice fed a high fat diet for 12 weeks after overnight fasting and oral fat tolerance testing with olive oil gavage. **D.** Cholesterol and triglycerides from FPLC fractions of pooled plasma from the 7-hour timepoint from the olive oil gavage in (C). **E.** Plasma TGs (left) and areas under the curve (right) after olive oil gavage in EL WT mice treated with Null adeno-associated virus vector (WT-AAV Null), *Lipg*$^{-/-}$ mice treated with Null vector (*Lipg*$^{-/-}$AAV Null), or *Lipg*$^{-/-}$ mice treated with AAV expressing murine *Lipg* directed to the liver (*Lipg*$^{-/-}$AAV WT). A left, C left, E left: P<0.05, repeated factor 2-way ANOVA, A right, C right, E right: *P<0.05, **P<0.01, student's unpaired T-test. Data is expressed as mean ± S.E.M.

understand the consequences of EL deficiency on impaired postprandial TRL clearance in the HFD-fed state, we performed lipidomics of plasma TG and phospholipids from HFD fed *Lipg*$^{-/-}$ vs WT mice after fasting and post-OFTT (**Fig 6A–6D, S1 and S2 Tables**). Comparing the levels of TG (**Fig 6A and 6B**) and PC (**Fig 6C and 6D**) species in plasma from fasted vs 7 hours post-OFTT treated mice, we found multiple species of each lipid specifically enriched in the plasma of *Lipg*$^{-/-}$ mice after OFTT relative to the WT mice. TG and PC species with polyunsaturated FAs were notably increased in the *Lipg*$^{-/-}$ plasma after OFTT, suggesting that not only PCs but also TGs enriched in PUFAs were less efficiently cleared from the circulation in the absence of EL (**Fig 6B and 6D**). This supports the finding of elevated serum levels of phospholipids (**Fig 2B**) and specifically multiple PUFAs in human *LIPG* p.Asn396Ser carriers from NMR metabolomic analyses described above.

While EL has been shown to be predominantly a phospholipase [14], it does also demonstrate some TG lipase activity *in vitro*. Our above data suggested that loss of EL activity delays lipolysis of long-chain PUFAs in both TGs and PCs. We sought to better understand the relative contributions of EL phospholipase and triglyceride lipase activities to TRL lipolysis in

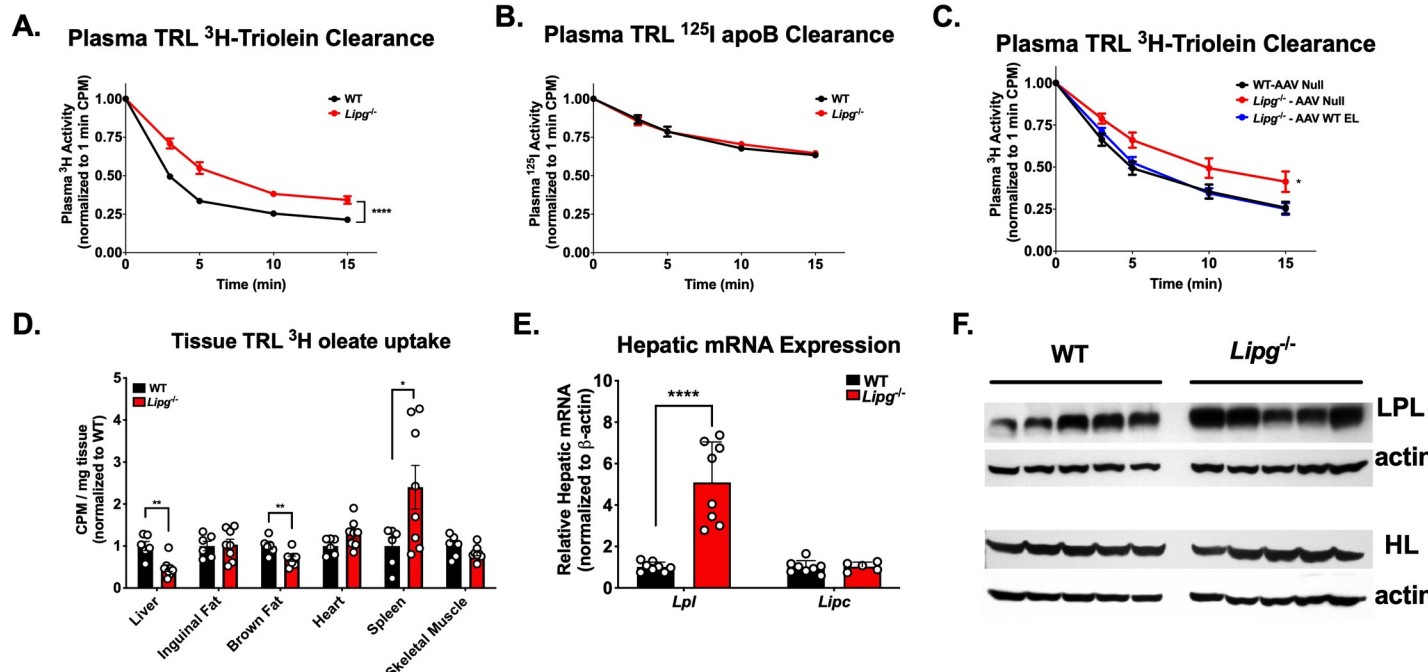

**Fig 5.** *Lipg* deficiency delays TRL lipolysis in mice. **A.** [3]H oleate plasma clearance curves from EL WT vs. *Lipg*[-/-] mice administered [3]H-triolein labeled human TRLs. Mice were administered radiolabeled TRLs and plasma [3]H levels were measured over 15 minutes and normalized to the 1 minute timepoint. **B.** Clearance of [125]I tyramine cellobiose labeled human TRLs in EL WT vs. *Lipg*[-/-] mice fed a high fat diet for 12 weeks. Plasma clearance of [125]I-TRL apolipoprotein B (apoB) after intravenous TRL administration was measured over 15 minutes. **C.** [3]H-triolein-labeled TRL clearance in WT mice treated with AAV Null vector, *Lipg*[-/-] mice treated with AAV Null, or *Lipg*[-/-] mice treated with AAV expressing murine *Lipg* and all fed a high-fat diet for 12 weeks and then treated with radiolabeled TRLs as described in (A). **D.** Uptake of [3]H-oleate into the indicated tissues after 15 min of administration of [3]H-triolein labeled TRLs in the mice from (A). [3]H activity in a fixed amount of tissue was normalized to the 1 minute plasma [3]H activity for each mouse. Normalized tissue [3]H activity is expressed for each tissue relative to the mean of the WT group. **E.** Hepatic mRNA expression of *Lpl* and *Lipc* in mice from (A). Quantitative real-time PCR cycle number for each gene was normalized to that of β-actin. **F.** Immunoblots of LPL and HL from liver lysates of mice in (A). Immunoblots of the proteins from the lysates for β-actin are shown as a loading control. A, C: *P<0.05, ****P<0.0001, repeated factor 2-way ANOVA. D & E: *P<0.05, **P<0.01, ****P<0.0001, student's unpaired T-test. Data is expressed as mean ± S.E.M.

concert with the predominant TG lipase LPL. We generated dual [3]H-triolein TG and [14]C-DPPC PC labeled lipid emulsions and compared TG and phospholipase activities of the two enzymes. As previously demonstrated [14], LPL demonstrated substantially greater TG lipase activity against TRL-like emulsions than WT EL, which demonstrated a modest but dose-dependent contribution to TG lipolysis (**Fig 6E**). Likewise, WT EL demonstrated a significant dose-dependent phospholipase activity on TRL-like emulsions compared to LPL (**Fig 6F**). The combination of WT EL with LPL substantially increased PC lipolysis on TRL-like emulsions relative to that of LPL alone (**Fig 6G**) and also increased TG lipolysis above that of LPL alone (**Fig 6H**). A catalytically inactive dominant negative variant p.Ser169Ala that we previously showed to reduce WT murine EL secretion from COS7 cells in vitro and delay LDL-PC catabolism in vivo when overexpressed [22] was also tested (**Fig 6E–6H**). This variant demonstrated markedly impaired phospholipase activity on TRL-like emulsions compared with WT EL when added alone (**Fig 6F**) or in combination with LPL (**Fig 6G**). EL p.Ser169Ala also impaired TG lipolysis when incubated alone (**Fig 6E**) or in the presence of LPL (**Fig 6H**) compared with WT EL which promoted TG lipolysis in both settings (**Fig 6E and 6H**).

Taken together, the above data suggest that the lipolytic activity of EL promotes TG lipolysis in addition to PC lipolysis on TRLs. This model was further supported by TG lipase activity assays using [3]H triolein labeled isolated human VLDLs in which assays were carried out for 4 hours after incubation with WT EL alone, LPL alone or EL and LPL combined in each molar

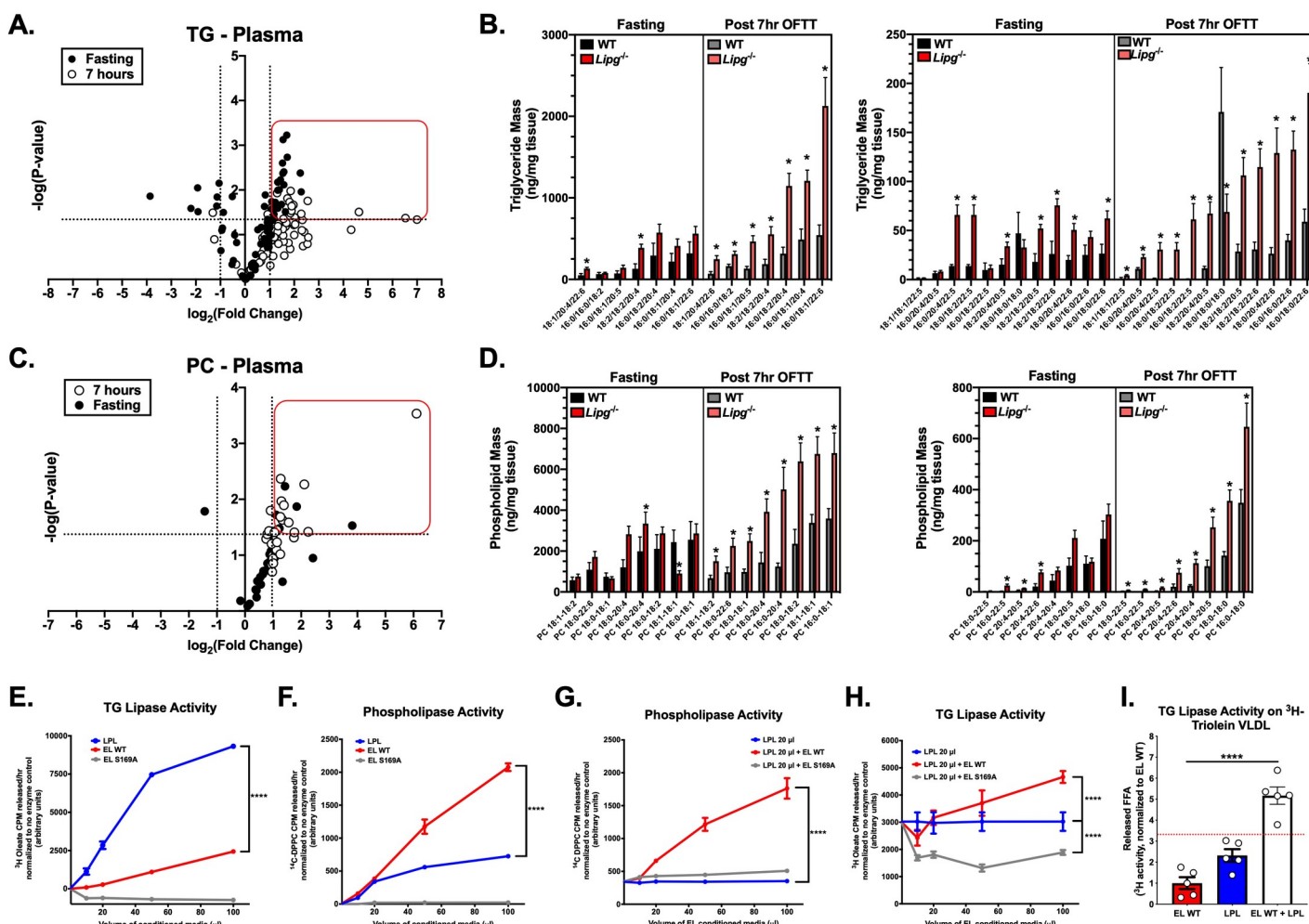

**Fig 6. EL and LPL cooperate to promote TRL lipolysis *in vivo*. A.** Volcano plot of 77 plasma triglyceride (TG) species from plasma of WT vs *Lipg*⁻/⁻ mice after overnight fasting (filled black circles) and 7 hours (white circles) after oral fat tolerance test. Lipids were measured by liquid-chromatography tandem mass spectrometry (LC-MS) as described in the Materials and Methods. Plot shows fold change in *Lipg*⁻/⁻ mice vs WT (x-axis; logarithmic base-2 scale) with the p-value (y-axis; logarithmic base-10 scale). Vertical dotted lines indicate two-fold change, and horizontal dotted line corresponds to P<0.05. **B.** (left and right) Plasma concentration of TG species in the red line from (A) as measured by LC-MS at the indicated timepoints. **C.** Volcano plot of 28 plasma phosphatidylcholine (PC) species from plasma of WT vs *Lipg*⁻/⁻ mice after overnight fasting (filled black circles) and 7 hours (white circles) after oral fat tolerance test. Lipids were measured by liquid-chromatography tandem mass spectrometry (LC-MS) as described in the Materials and Methods. **D.** (left and right) Plasma concentration of PC species in the red line from (C) as measured by LC-MS at the indicated timepoints. **E.** In vitro TG lipase activity of EL and LPL conditioned media on ³H-triolein labeled TRL-like emulsions. Emulsions were incubated with the indicated volumes of EL or LPL-containing conditioned medium from COS7 cells infected with EL- or LPL-expressing adenovirus. Release of ³H-oleate was used to measure TG lipase activity. **F.** In vitro phospholipase activity on ³H-triolein labeled TRL-like emulsions in reactions with LPL of the indicated volume and increasing volumes of EL WT or EL S169A conditioned media. **G.** In vitro phospholipase activity of EL and LPL conditioned media on ¹⁴C-DPPC-labeled TRL-like emulsions, as shown in (A-B). **H.** TG-lipase activity on ¹⁴C-DPPC labeled TRL-like emulsions in reactions with LPL of the indicated volume and increasing volumes of EL WT or EL S169A conditioned media. **I.** TG-lipase activity on ³H-triolein labeled human VLDLs incubated with recombinant EL WT, LPL or a combination of EL WT and LPL. Red dotted line indicates the sum of the mean activities for the EL WT (red bar) and LPL (blue bar) groups. B and C: *P<0.05, *student's unpaired T-test comparing the indicated groups. E-H: ****P<0.0001, two-way ANOVA for interaction term of indicated groups and sample volume term (X-axis). I: ****P<0.0001, one-way ANOVA comparing EL WT + LPL group (white bar) with other groups. Data is expressed as mean ± S.E.M.

ratios. We found that the relative increase in the amount of TG lipolysis with EL and LPL combined was greater than the sum of the amount of TG lipolysis from each enzyme individually (red dotted line representing sum of means for EL WT and LPL groups (~3.32), compared with EL WT + LPL group (~5.17), as seen in **Fig 6I**).

To assess whether EL and LPL may similarly contribute collaboratively to TG lipolysis in vivo, we somatic hepatic overexpression of the circulating LPL inhibitor ANGPTL4 [35] to

inhibit LPL. We used an AAV vector expressing murine ANGPTL4 in livers of WT or *Lipg*[-/-] mice fed a chow diet and compared plasma lipids over 6 weeks after AAV expression and postprandial TGs after OFTT (**Fig 7**). We found as expected that *Lipg*[-/-] mice expressing ANGPTL4 had increased fasting TGs compared to *Lipg*[-/-] mice injected with a control AAV due to inhibition of LPL. More importantly, *Lipg*[-/-] mice expressing ANGPTL4 had significantly higher fasting TGs than WT mice expressing ANGPTL4 AAV (**Fig 7D**). Seven hours after OFTT, we found that the *Lipg*[-/-] mice expressing ANGPTL4 had markedly elevated TGs (**Fig 7E**). The relative increase in postprandial TGs in this group (~218%) was notably greater than the sum of the increases in the *Lipg*[-/-]—AAV Null (lacking EL only) and WT—AAV ANGPTL4 (lacking LPL only) groups combined (~140%, indicated by red dotted line, **Fig 7F**). Thus combined loss of both EL and LPL activity has a profound effect in impairing TG lipolysis in vivo. Taken together, these in vitro and in vivo data support a model in which EL collaborates with LPL to promote TRL TG lipolysis.

## Discussion

Since the first reports of the cloning of *LIPG* in 1999 [10, 11], its encoded protein EL emerged as a critical regulator of HDL catabolism in animal models and humans [36, 37]. More recently, EL has also been suggested to have a potential role in LDL metabolism in vivo [22–24], presumably through its ability to promote LDL phospholipid hydrolysis. However, EL has not been previously shown to have a physiological role in TRL metabolism. Here we show that both humans and mice with genetic loss-of-function of EL have substantially reduced ability to hydrolyze TRL-TGs. We show that the absence of EL in mice delays TRL-TG lipolysis, TRL clearance and that these effects are exacerbated by increased dietary or postprandial TGs. We demonstrate that EL deficiency delays release of long chain PUFAs from TGs and PCs postprandially. We show that *in vitro* EL when combined with LPL promotes greater TG lipolysis of human VLDLs than does LPL alone. Furthermore, we show that *in vivo* the combined loss of both EL and LPL activity impairs TRL metabolism to an even greater extent than does loss of LPL alone. We suggest that EL contributes collaboratively to the remodeling of TRLs by LPL and allows more robust TG lipolysis by LPL, ultimately promoting more efficient FA uptake in peripheral tissues and hepatic TRL clearance (**Fig 8A**). In the absence of EL, our data indicates that there is less efficient TRL-TG lipolysis and a delay in TRL clearance (**Fig 8B**).

Our work demonstrates that both human and murine germline loss-of-function of *LIPG* contribute to an underappreciated role of the encoded enzyme EL in TRL catabolism. The first studies of human genetics of *LIPG* in humans clearly established that genetically reduced EL activity causes elevations in HDL-C [19, 20, 38–40]. These included the p.Asn396Ser (rs77960347) studied here that was subsequently employed as a genetic instrument to ascertain a nonsignificant relationship between genetically elevated HDL-C and risk of myocardial infarction via Mendelian randomization [27]. More recently, very large studies [29, 30] have clearly established that this variant is significantly associated with elevated LDL-C as well, consistent with data in mice indicating a role for EL in LDL metabolism. Our study provides a comprehensive demonstration that the reduced function p.Asn396Ser variant is also associated with elevated TGs in humans. The relatively modest association between genetic *LIPG* loss-of-function and TGs relative to the relationship seen with HDL-C and LDL-C may reflect the lower phospholipid content of TRLs relative to these other lipoprotein classes, and variability in TGs in fasted vs. nonfasted states, for which inclusion differed among the UKBB, GLGC, and MVP studies. While more powered studies are required to refine the association of additional genetic variants in *LIPG* with measures of TRL metabolism, our findings here support the notion that *LIPG* plays analogous roles in TRL metabolism across humans and mice. An

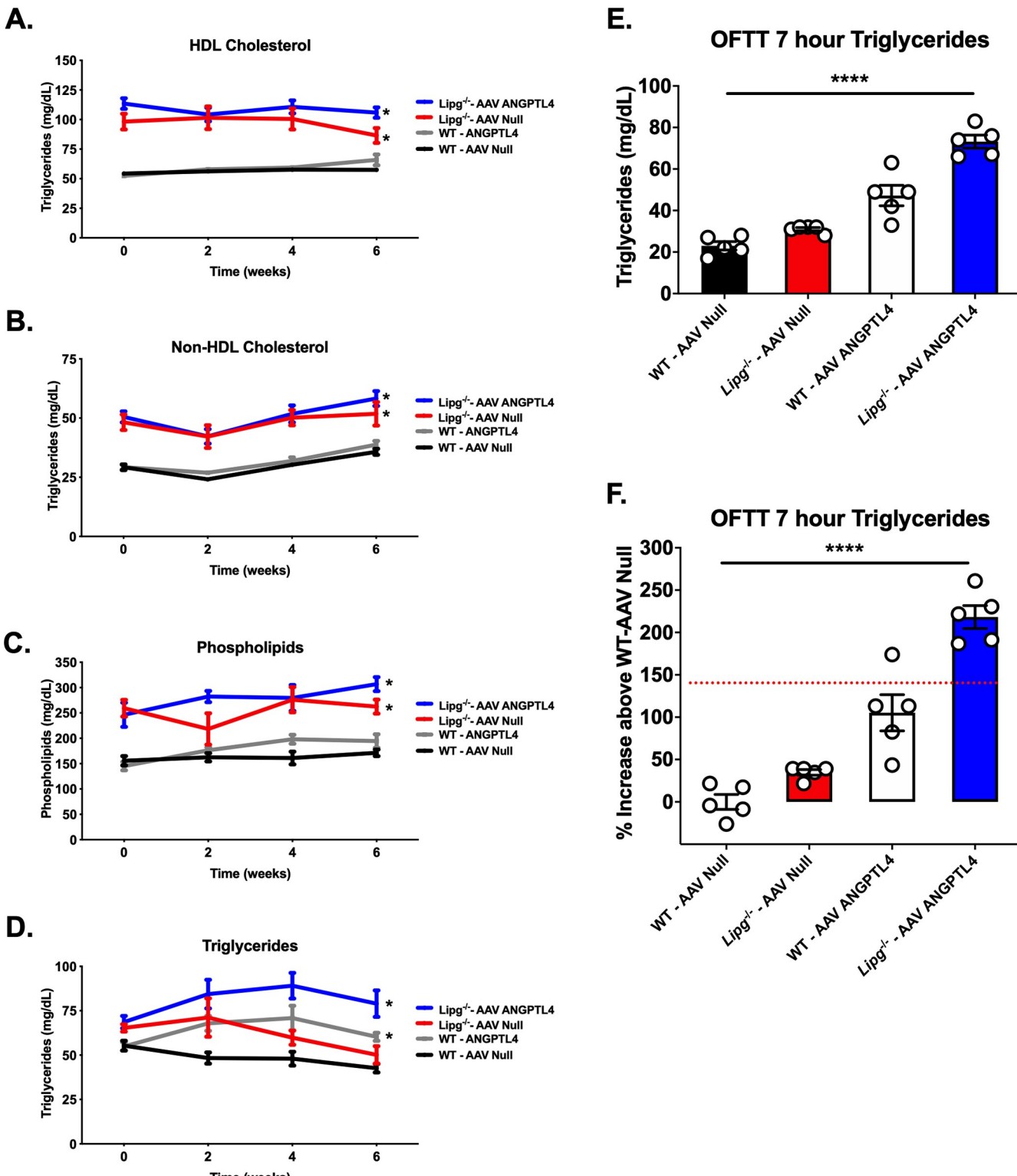

**Fig 7. Impact of LPL inhibition with ANGPTL4 overexpression in *Lipg*<sup>-/-</sup> mice. A-D.** Plasma HDL-C, nonHDL-C, phospholipids, and TGs from WT vs. *Lipg*<sup>-/-</sup> mice treated with AAV Null vector (WT-AAV Null; *Lipg*<sup>-/-</sup>—AAV Null) or AAV ANGPTL4 vector (WT-AAV ANGPTL4; *Lipg*<sup>-/-</sup>—AAV ANGPTL4) after a 4 hour fast. Lipids were measured by chemical autoanalyzer. **E-F.** Plasma TGs 7 hours after oral fat tolerance testing in mice from (A). TGs in mg/dl are shown in (E). The relative difference in percent (%) increase above the WT–AAV Null group in plasma TGs from each group in (E) is shown in (F). In (F) the red dotted line shows the mean summed % increase in 7 hour plasma TGs for the *Lipg*<sup>-/-</sup>—AAV Null and the WT-AAV ANGPTL4 group. For A-D,

*P<0.05, **P<0.01, ***P<0.001, two-way ANOVA comparing the indicated groups to the WT AAV-Null group. For E and F, ****P<0.0001, two-way ANOVA across groups comparing interaction of genotype (WT vs *Lipg*<sup>-/-</sup>) and AAV (Null vs ANGPTL4). All error bars indicate mean ± S.E.M.

investigation in a subset of ~50,000 individuals of the UK BioBank for which both LIPG and LPL pLOF variants were annotated did not yield discovery of any human carriers of combined lipase genetic loss-of-function. By exploring the phenotypes of *Lipg* deficiency and combined lipase loss-of-function further in mice, our study demonstrates the utility of comparative studies of gene loss-of-function in humans and experimental models to assess the contribution of a gene function to a new phenotype.

Our data here unveils EL as an important contributor to the complex biology of extracellular TG lipolysis. Indeed, murine physiology and human genetics have supported the notion that activation of the LPL pathway, through LPL itself and its promoters GPIHBP1, apoC-II and apoA-V, or inactivation of its inhibitors apoC-III, ANGPTL3 and ANGPTL4, is the central axis governing TRL clearance [1]. Here we demonstrate that EL is also necessary for efficient TRL TG lipolysis and functions in concert with LPL in mice.

The mechanisms by which EL contributes to TRL metabolism may include both its phospholipase activity, which is its predominant lipase activity, as well as an underappreciated direct TG lipolysis. Here we show in our in vitro experiments and in lipidomic characterization of postprandial TRLs from *Lipg*<sup>-/-</sup> mice that loss-of-function of EL impairs both phospholipid and TG lipolysis on TRLs. Since TRLs possess a phospholipid shell surrounding a neutral lipid TG core and our data and prior work suggests that EL possesses significantly higher phospholipase:TG lipase activity on a spectrum of lipoproteins, we hypothesize that one mechanism by which EL confers more efficient TG lipolysis along with LPL is through "unshielding" the

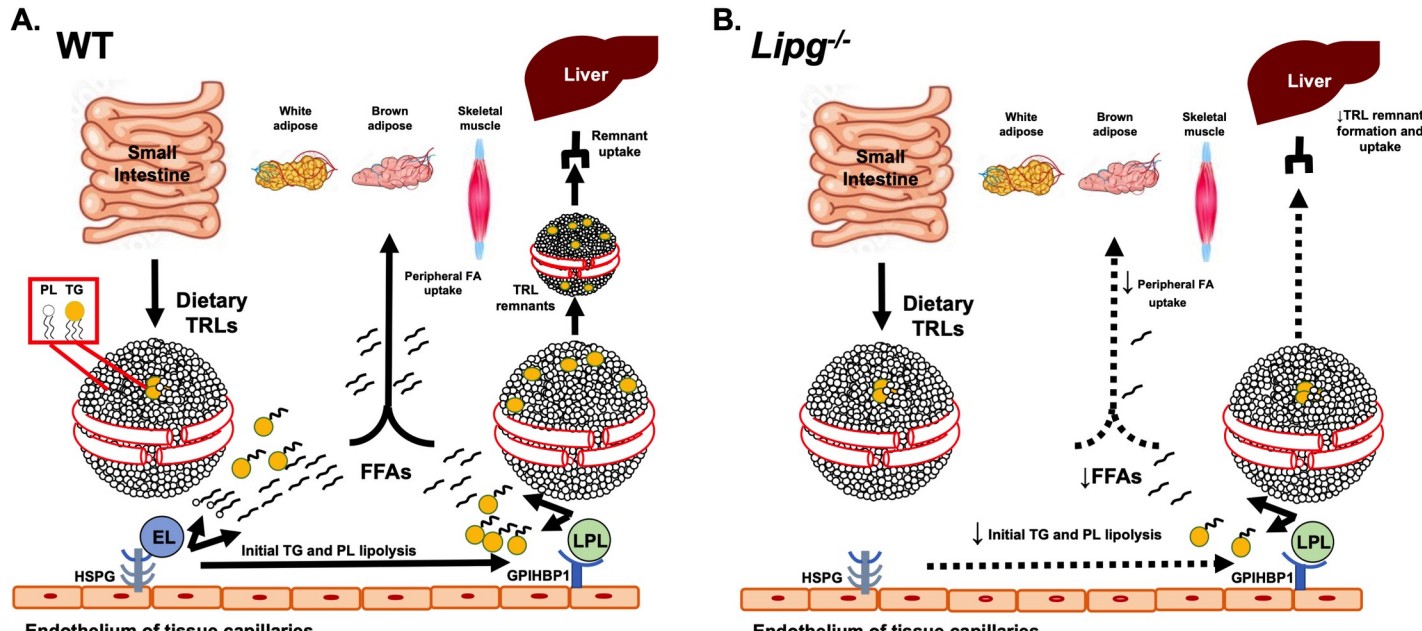

**Fig 8. Schematic of role of EL in TRL catabolism. A-B.** In the setting of EL expression (A), phospholipid lipolysis on TRL in postprandial or high fat-fed state allows for synergistic and efficient TG lipolysis by LPL, which promotes FA and TRL remnant uptake by the liver. Multiple FAs including long chain PUFAs are internalized by the liver. In EL-deficient mice (B), lipolysis of phospholipids on the outer shell of TRLs is impaired, resulting in less efficient access of LPL to TRL TGs and reduced TG lipolysis and remodeling of TRLs to smaller remnants, thus reducing TRL remnant uptake by the liver. In addition, the lack of ability of EL to catalyze lipolysis of phospholipids containing its preferred substrates of long chain PUFAs results in decreased PUFA uptake by the liver.

exterior PCs through its phospholipase activity and allowing greater access by LPL to the TGs in the lipoprotein core. Our data on the p.Ser169Ala variant in EL also demonstrated that this catalytically inactive variant blunted both PC and TG lipolysis compared with WT EL, suggesting that the lipolytic activity of EL is an important contributor to TRL catabolism. This variant also demonstrated an ability to reduce overall TG lipolysis in the presence of robust lipolysis from LPL but without WT EL present. The mechanism behind this observation remains unclear from the current data. Prior work on this variant has suggested that it may have a dominant negative function in impairing endogenous murine EL secretion when co-overexpressed in COS7 cells [22]. It remains possible as previously speculated that a nonlipolytic function related to ligand binding may be impacted by this variant [41], which could explain why it may also impact TG lipolysis in the presence of LPL even when WT EL is not present for the variant to inhibit. Overall our data presented here is limited in its ability to dissect the relative contributions of phospholipase, TG lipase and any nonlipolytic activities of EL to determine which of these functions predominates in the ability of EL to promote TRL-TG lipolysis. Our work does demonstrate that EL has underappreciated enzymatic TG lipase activity on TRLs and that this likely contributes at least in part to the markedly enhanced TRL lipolysis seen in concert with LPL in vitro and in vivo.

In addition to its role in TG-rich lipoprotein catabolism, our findings her demonstrate both through the human genetics and in mice that EL is critical for LDL metabolism as well. We demonstrated through our murine loss-of-function model and marked increase in LDL-C in $Lipg^{-/-}$ mice particularly after high fat diet feeding. This extends prior work from our laboratory that demonstrated LDL as a biochemical substrate for EL-mediated phospholipid lipolysis in vitro [14, 21] and further support from our group through viral vector mediated expression of *LIPG* in mice showing that EL catabolizes LDL-phospholipids in vivo to promote LDL turnover [22]. That prior work demonstrated that EL overexpression in multiple models of increased circulating LDL (LDLR-deficiency, apoE deficiency, or transgenic overexpression of apolipoprotein B) increased LDL catabolism, and our findings here in $Lipg^{-/-}$ mice in which apoB-containing lipoproteins were increased through dietary cholesterol and saturated fat elevation are complementary to that prior work. Consistent with this, another group examining apoB-containing lipoprotein catabolism in rabbits with transgenic overexpression of human *LIPG* observed increased apoB-containing lipoprotein turnover through increased EL-mediated phospholipase activity [42]. More recently, we and others have shown that EL may mediate a LDL receptor-independent pathway for LDL clearance in vivo in a manner that is "masked" by ANGPTL3 inhibition of EL and could be derepressed through antibody-mediated or siRNA-mediated inhibition of ANGPTL3 [23, 24]. These data support the notion that EL is a critical regulator of LDL-C catabolism. Our data here extends the notion that EL facilitates further remodeling of a spectrum of apoB containing lipoproteins including large TRLs to allow more efficient TG lipolysis by LPL and thus physiologically promotes clearance in vivo.

Collaboration between EL and LPL to promote TRL clearance in nutrient-rich states has been suggested through demonstration of partial reciprocity of these lipases in the setting of LPL deficiency in vivo. Mice lacking LPL globally exhibited upregulated EL expression and activity in white adipose tissue, commensurate with increased HDL-phosphatidylcholine (PC) catabolism and incorporation of these FAs in adipose lipid droplets [25]. A similar finding was observed in GPIHBP1-deficient mice lacking the critical translocation scaffold for LPL to exist on the luminal surface of capillary endothelial beds [43]. This work suggested a reciprocal relationship between LPL activity and EL activity in tissues that primarily serve to liberate FAs from TRLs, suggesting that EL activity could serve as an adaptation to partially overcome the lack of LPL activity that would otherwise deprive critical metabolic tissues of lipoprotein-associated fatty acids.

Our data show that multiple species of PCs and TGs with long chain PUFAs accumulate in the postprandial circulation of HFD-fed mice deficient in EL. This suggests that EL may be an important conserved mediator of PUFA metabolism. Given our finding that systemic deficiency of EL not only impairs clearance of these PUFAs but also decreases hepatic FA uptake, we posit that EL may serve physiologically as a key generator of these PUFAs for the liver. The role of this function remains unclear but it may be that PUFA generation by EL in the liver may serve protective roles in nutrient-rich states, such as protection from lipogenic gene expression as has been suggested previously [33, 44]. The physiological consequences of EL-mediated lipolysis of TRLs and liberation of PUFAs remains underexplored and an important area for further investigation.

In summary, we examined the contribution of EL and its phospholipase activity to TRL catabolism through comparative studies of loss-of-function in humans and mice. We show that EL plays a physiological role in mediating postprandial TRL TG lipolysis and clearance. We demonstrate the physiological cooperativity between EL and LPL in the sequential remodeling of circulating TRLs through phospholipid and TG lipolysis. Our data positions EL as a key node in the complex process of efficient lipolysis of TRLs, a phenomenon of clinical importance of myriad cardio-metabolic diseases.

## Materials and methods

### Ethics statement

Mouse husbandry and experiments were reviewed and approved by the University of Pennsylvania Institutional Animal Care and Use Committee (IACUC; protocol number 803056).

### Human LIPG p.Asn396Ser genetic variant association with plasma lipid traits

We mined human genetic consortia performing exome-wide association studies of plasma lipid traits for association of the *LIPG* p.Asn396Ser variant (rs77960347), a low frequency non-synonymous coding variant previously reported to confer loss-of-function to EL in humans and reduced phospholipase activity in vitro and in vivo [29, 45]. The following three cohorts were assessed: UK BioBank (UKBB), Global Lipids Genetics Consortium (GLGC), and the Million Veteran Program (MVP). Informed consent was obtained for participants for all three studies.

The UKBB is a population based BioBank comprising ~500,000 individuals aged 39–72 years old living in the United Kingdom who attended one of 21 centers in the UK with blood sampled for genetic, clinical and biomarker analysis along with anthropometric trait measurements including weight, height and body mass index [28, 46]. Of this cohort, ~370,000 individuals undergoing genome-wide genotyping [46] and ~200,000 individuals undergoing whole-exome sequencing as previously described [28] were included in the analysis of the LIPG p. Asn396Ser variant. Genome annotation was completed using the GRCh38 assembly with 20X coverage at 95.6% of sites on average. Study protocol and data access and details of the exome sequencing variant calling and quality control are available upon request online [28]. Plasma lipid measures (HDL-C, LDL-C, total cholesterol, triglycerides) were measured directly for all participants between 2006–2010 in a nonfasting state using a chemical auto-analyzer. Lipid measures (TG, total cholesterol, HDL-C, LDL-C) were natural-log transformed. For each lipid trait, effect estimates were obtained after controlling for sex, age, age-squared, and up to the first four principal components using linear mixed models. Effect estimates are reported in SD units

The Global Lipids Genetics Consortium (GLGC) study is an exome-wide genotyping association study in ~300,000 predominantly European individuals [29]. Effect estimates for the association of *LIPG* p.Asn396Ser with HDL-C, LDL-C and TGs were derived from the reported summary statistics for the referenced GLGC association study at http://csg.sph.umich.edu/willer/public/lipids2017/. Details of the study populations, genotyping and analysis can be found in the associated study [29]. Briefly, fasting lipid measures were used exclusively for LDL-C and TG analyses. LDL-C was calculated using the Friedewald equation for subjects with TG < 400 mg/dl (LDL-C = TC–HDL-C–(TG/5)). Lipid measures were natural-log transformed A maximum of 237,050 European participants was included for any single-variant analysis in the association study. For each lipid trait, effect estimates were obtained after controlling for sex, age, age-squared, and up to the first four principal components using linear mixed models as described, using the program RAREMETALWORKER or RVTESTS. Effect estimates are reported in SD units.

The Million Veteran Program (MVP) is a multi-center study of participants aged 19–104 from more than 50 VA Medical Centers across the USA since 2011 with inclusion of electronic health record data, International Classification of Diseases (ICD-9) diagnosis codes, clinical laboratory measures and samples for human genetic analysis [30]. Whole-exome sequencing of DNA from participants was performed and variant quality control completed as previously described. Participants were grouped into three mutually exclusive ethnic groups for analysis: non-Hispanic whites (European ancestry), non-Hispanic blacks (African ancestry), and Hispanics. Plasma lipid measures from electronic health records for participants was extracted as the maximum measure of LDL-C, TG, total cholesterol and minimum HDL-C for each participant, so as to estimate plasma lipid concentrations in the absence of lipid lowering treatment. Lipid measures were natural-log transformed Effect estimates were obtained after adjusting for age, age$^2$, sex and 10 principal components of ancestry. Further details of quality control of the lipid measures from this study are as previously described [30]. In addition to separate analyses this variant in the UKBB, GLGC, and MVP cohorts, we also pooled effect estimates for lipid traits across cohorts and performed a fixed-effects meta-analysis using the meta package in R (version 3.6.0).

## Association of LIPG p.Asn396Ser variant with serum lipid and lipoprotein metabolites by NMR

We examined a prior genome-wide association study of serum metabolites measured by NMR comprising up to 24,925 European participants as previously described [31]. Informed consent was obtained from all participants. Up to 123 serum metabolites were measured from participants as previously reported [31]. We considered a subset of 84 serum lipid and lipoprotein-related metabolites. Genotyping was performed with SNPs imputed up to 39 million markers using the 1000 Genomes Project March 2012 release with human genome build 39. Effect estimates were obtained after adjusting for age, age$^2$, sex and 10 principal components of ancestry. Given the correlation between circulating levels of lipoprotein-related traits, we considered FDR q <0.05 as the experiment-wide significance threshold.

## Animals

C57BL/6 (or wild-type, WT) mice were obtained from Jackson Laboratories (stock number 000664). Generation of endothelial lipase (EL) knockout mice (*Lipg*$^{-/-}$) on a C57BL/6 background were previously described [17]. *Lipg*$^{-/-}$ mice were crossed at least four generations with mice on the C57BL/6 background. All mice were maintained on a normal chow diet and a 12 hr light/12 hr dark cycle. All studies were approved by the University of Pennsylvania

Institutional Animal Care and Use Committee (IACUC, protocol number 803056). Comparisons of fasting plasma lipids on chow diet were done with age- and sex-matched mice of each genotype. For experiments involving high-fat diet feeding, 10–12 week old mice were fed a high fat diet (D12451—Open Source Diets—45% Kcal from fat) for the indicated durations. Such experiments were initially performed in both male and female mice for which fasting plasma lipids were measured every four weeks after initiation of high-fat diet feeding. Subsequent experiments were performed in age-matched male mice. Data shown in the manuscript represents experiments with male mice.

Blood from the mice was collected either via retro-orbital bleeding under isoflurane anesthesia, using EDTA containing-capillary tubes. For euthanasia, mice were sacrificed by cervical dislocation under deep isoflurane anesthesia. Lipid measurements were performed on mouse plasma using an Axcel autoanalyzer. In addition, for some experiments, plasma was pooled by experimental group freshly after collection and 150 µl of plasma was separated by fast-protein liquid chromatography (FPLC) on a Superose 6 gel-filtration column (GE Healthcare Life Sciences, Pittsburgh, PA, USA) into fractions of 0.5 ml volume each. Total cholesterol and TG were measured from FPLC-separated fractions using Infinity Liquid Stable cholesterol and triglyceride reagents (Thermo Scientific, Waltham, MA, USA) in 96-well microplates using a Synergy Multi-Mode Microplate Reader (BioTek, Winooski, VT, USA).

## Oral Fat Tolerance Test (OFTT)

In order to determine post-prandial TG clearance in mice, oral fat tolerance testing (OFTT) was performed as previously described [47, 48]. Briefly, WT and *Lipg*$^{-/-}$ mice on high fat diet or chow (n = 5–8 per group) were fasted overnight for 14 hours. Mice for these experiments were 10–12 weeks old (Fig 4A), 22–24 weeks old (after feeding high fat diet for 12 weeks; Fig 4C), or 6–8 weeks old (Fig 7). During this period mice were maintained at 24˚C under a normal 12 h light–dark cycle and had access to water. Prior to gavage, blood was collected to determine the baseline TG plasma levels. Mice were then weighed and gavaged with 10 µL/g of fasting body weight of olive oil (Sigma Aldrich). After gavage, blood was collected at 1, 3, 5 and 7 hours via retro-orbital bleeding, using EDTA-containing capillary tubes. Blood was spun and plasma was collected for TG determination and FPLC. For these experiments, plasma TG content was measured by colorimetric enzymatic assays in a 96-well microplates (Infinity Triglycerides reagent, Thermo Scientific, Waltham, MA, USA). Where indicated, FPLC-fractionation of pooled plasma from mice at the 7 hour timepoint from OFTT experiments was done as described above.

## Intravenous radioisotope-labeled TRL clearance

Human TG-rich lipoproteins (TRL) were purified by sequential ultracentrifugation of the plasma of non-fasted healthy volunteers. Plasma was isolated from 200 ml of blood. It was then transferred to ultracentrifugation tubes and a layer of KBr solution (density = 1.006 g/L) was added on top. The tubes were sealed and ultracentrifugated in a Beckman XL590 Ultracentrifuge (Beckman Coulter) using a 70.1 Ti rotor (Beckman Coulter) at a speed of 40,000 RPM for 18 hours. At the end of the run, TG-rich lipoproteins, including VLDL, CM and remnants, that are characterized by a density <1.006 g/L floated on the top of the tube. Lipoproteins were then collected and the total protein content was measured by BCA assay (Pierce BCA Protein Assay Kit, Thermo Scientific) and further processed for radioactive labeling.

For $^3$H-Triolein labeling of TRLs, isolated human VLDLs were labeled with $^3$H-Triolein (TO) as previously described [48]. Briefly, TRLs (containing 3 mg protein) was labeled with 0.5 mCi of $^3$H-TO (Perkin Elmer). 0.5 mCi of $^3$H-TO in toluene was dried down under a

nitrogen and resolubilized in 150 μl of ethanol. $^3$H-TO solution and ultracentrifugally-isolated lipoprotein deficient human plasma (100 mg protein / 3 mg TRL protein) were added to the TRLs and the resulting mixture was incubated at 37˚C overnight. Labeled TRLs were then subjected to a "wash" ultracentrifugation step under the same conditions described above to re-isolate the TRLs and remove excess non-incorporated $^3$H-TO. TRLs were then collected, dialyzed against PBS, and fractionated by FPLC to measure TG and $^3$H-TO activity in the VLDL-CM fractions. $^3$H activity was determined by scintillation counting (Beckman LS 6500 Scintillation System).

For $^{125}$I-protein labeling, dialyzed human TRLs were labeled with $^{125}$I by the iodine monochloride method as previously described [48]. Lipoproteins were then dialyzed against PBS to remove excess of non- incorporated $^{125}$I. For direct $^{125}$I labeling, 1.5 mg of TRL in PBS were iodinated with 1 mCi of $^{125}$I, 300 μl of 1 M glycine, and 150 μl of 1.84 M NaCl / 2.84 μM ICl solution, vortexed and applied to a PG510 desalting column (Amersham Biosciences) that was pre-equilibrated with 0.15 M NaCl / 1 mM EDTA solution. Iodinated proteins were eluted in NaCl/EDTA solution to a final volume of 2 mL. They were then dialyzed against PBS and protein concentration was assessed by BCA assay. $^{125}$I activity was determined by gamma counting (Packard Cobra II Auto5 Gamma counter).

*In vivo* experiments were designed to determine the plasma lipolysis and particle clearance rates of TRLs from the circulation. For each experiment, WT and *Lipg*$^{-/-}$ mice (n = 5–6 per group) were fasted 4 hours and bled to determine baseline lipid levels. They were then injected with dialyzed $^3$H or $^{125}$I labeled TRLs by intravenous tail vein injection. Mice were bled at 1, 3, 5, 10 and 15 minutes after TRL administration. Mice were euthanized at 15 minutes after injection as previously described and tissues collected. Plasma $^3$H or $^{125}$I activity at each timepoint were determined using scintillation counting and gamma counting, respectively (Beckman LS 6500 Scintillation System and Packard Cobra II Auto5 Gamma counter). The relative $^3$H activity remaining the in circulation was calculated by normalizing the activity from each timepoint by that of the 1 minute timepoint for each mouse after subtracting the $^{125}$I activity spillover in the $^3$H measurement from the $^3$H activity. The fractional catabolic rates were calculated with the WinSAAM program (University of Pennsylvania) by fitting a biexponential curve to the $^{125}$I and $^3$H counts normalized to the 1 minute time point. The differential uptake of labeled $^3$H oleate in each tissue was determined by scintillation counting of ~150 μg of homogenized tissue in PBS. The results were expressed as activity/ mg of tissue and normalized to the 1 minute time point.

## In vivo VLDL-TG secretion

Hepatic VLDL secretion was measured in WT and *Lipg*$^{-/-}$ mice (n = 6–8 per group). Briefly, mice were fasted for four hours and then administered the poloxamer P407 (25 mg in 0.4 ml PBS) by intraperitoneal injection to inhibit peripheral lipolysis, as previously described [49]. Mice were bled at 0, 60, 120, and 240 minutes after poloxamer 407 (P407) administration for plasma collection. Plasma TGs were measured at each timepoint with the Infinity Triglycerides reagent (Thermo Scientific) in 96 wells microplates. Zero, 60 and 240 minute plasma samples were used for calculating the TG secretion rate by linear regression.

## Immunoblotting

For immunoblotting of FPLC fractions, 150 μl of each fraction was dried down to a final volume of 20 μl using a centrifugal evaporator at 37˚C (Genevac, SP Scientific). For immunoblotting of liver lysates, frozen tissue was homogenized in RIPA buffer containing a complete set of protease inhibitors (Roche), using a high speed mechanical homogenizer (TissueLyser II,

Quiagen). The homogenate was spun at 15000 rpm. The supernatant was collected, protein content was determined by BCA assay (Pierce BCA Protein Assay Kit, Thermo Fisher Scientific) and 50 ug of total protein were loaded into the gel. All samples were combined with lithium dodecyl sulfate sample buffer and reducing agent (Thermo Fisher Scientific), according to the manufacturer's specifications and heated at 95˚C for 10 minutes to help denaturation and then loaded into gradient gels. All runs were conducted in presence of a pre-stained protein standard to verify the expected protein molecular weight (SeeBlue Plus2, Invitrogen). For other proteins, 4–12% gradient was used with 3-(N- morpholino)propanesulfonic acid (MOPS) running buffer (Bis-Tris Gels and MOPS SDS running buffer, Thermo Fisher Scientific). Gels were run in the XCell SureLock apparatus and transferred to nitrocellulose membrane (Biorad) using the XCell II Blot Module with SDS and methanol containing transfer buffer (transfer buffer with 20% methanol, Thermo Fisher Scientific). The membranes were blocked over-night in 5% fat free milk (Thermo Fisher Scientific) and then immuno-blotted against target proteins. For this purpose the following antibodies were used:

- LPL: primary antibody—rabbit anti-LPL (H-53; sc-32885; Santa Cruz); secondary antibody —anti-rabbit IgG-HRP (sc-2030; Santa Cruz).

- HL: primary antibody—rabbit anti HL (H-70; sc-21007; Santa Cruz); secondary antibody— anti-rabbit IgG-HRP (sc-2030; Santa Cruz).

- ß-actin: primary antibody–mouse anti actin (clone AC-15; A5441; Sigma Aldrich); secondary antibody—anti mouse IgG-HRP (sc-358914), Santa Cruz.

Detection was performed by incubation of immunoblotted membranes with Luminata Crescendo Western HRP substrate (EMD Millipore) and the image was obtained using ChemiDoc Touch Gel and Western Blot Imaging System and analyzed using Image Lab software (Biorad).

## Adeno-associated virus vectors

An adeno-associated virus (AAV) vector encoding WT human *LIPG* (gene product EL) was created by cloning the coding region from an EL expression plasmid described previously[20] into the pAAV.TBG cis-plasmid [50]. An AAV vector encoding murine ANGPTL4 was created by cloning from an expression plasma encoding mouse ANGPTL4 into the pAAV.TBG cis plasmid as previously described [51]. The AAV2/8 vectors encoding EL and ANGPTL4 were produced by transfection into 293 cells, with an adenovirus helper plasmid and a chimeric packaging construct in which the AAV2 *rep* gene is fused with the *cap* gene of serotype 8. A control *LacZ* gene was packaged into an AAV, also of serotype 2/8. Genome copy was determined by TaqMan (Applied Biosystems) analysis as previously described [48]. For the indicated experiments involving AAV transgene expression in WT vs. *Lipg*[-/-] mice, $3 \times 10^{11}$ genome copies of virus as determined by gene expression analysis were administered by intraperitoneal injection into groups of mice (n = 6–8 per group). Viruses were administered in 0.9% phosphate-buffered saline vehicle under sterile conditions. For OFTT experiments in mice administered WT EL AAV, experiments were performed four weeks after AAV administration. For the OFTT experiment in mice administered ANGPTL4 AAV, experiments were performed 6 weeks after AAV administration. Control mice in these experiments were administered equivalent doses of Null AAV2/8 vector lacking a transgene, as previously described [20, 48].

## Lipidomics from mouse plasma

For plasma lipidomics by liquid chromatography-mass spectrometry (LC/MS), total lipids were first extracted from mouse plasma from WT vs *Lipg*[-/-] mice after 12 weeks of feeding a

high fat diet and fasted overnight or after 7 hours of OFTT (20 μl per sample). Total lipids were extracted using the Bligh and Dyer method [52]. Analysis was done using an AB Sciex QTRAP 5500 mass spectrometer coupled to an Agilent 1260 HPLC system as described previously [53]. The total lipid extract was mixed with internal standards (50 ng each) and separated on a normal phase column (Supelco Ascentis Si 3μ, 10cm × 2.1 mm). The solvent system used was: 100% solvent A (chloroform/ methanol/2 mM aqueous ammonium chloride, 80:19.5:0.5 by vol.) from 0 to 5 min; then a linear gradient of 100% solvent A to 100% solvent (chloroform/methanol/water/2 mM aqueous ammonium chloride, 60:34.5:5:0.5; by vol.) from 5 to 30 min; and 100% solvent B from 30 to 35 min. The flow rate was 0.35 ml/min. The ESI source temperature was set at 450˚C. Multiple reaction monitoring (MRM) was used to quantify the various PC and TG species. Data acquisition and processing were controlled using the Analyst 1.5 software (Applied Biosystems, Foster City, CA, USA).

## Hepatic gene expression

Immediately after euthanasia tissues were collected and snap frozen in liquid nitrogen to preserve nucleic acids. Total RNA was extracted from tissues using Trizol (Thermo Fisher). Tissues were homogenized in Trizol with a high speed mechanical homogenizer (TissueLyser II, Qiagen) and RNA was extracted with chloroform/isopropanol method according to the manufacturer specifications (Trizol, Thermo Fisher). RNA was quantified by spectrophotometric absorbance determination at 260 nm (Take3 Micro-Volume Plates. BioTek). Absorbance at 230 and 280 nm was also measured to determine nucleic acid purity. 0.1 μg of RNA was reverse transcribed into cDNA using High-capacity cDNA Reverse Transcription Kit (Applied Biosystem). Gene expression was evaluated through quantitative Real Time PCR, using pre optimized reagent and QuantStudio 7 Flex Real-Time PCR System (Applied Biosystems). Briefly, the reaction mixture included a mix of unlabeled PCR primers and a TaqMan probe corresponding to each gene measured with FAM dye label (Applied Biosystems—TaqMan Gene Expression Assays), and TaqMan Fast Advanced Master Mix (Applied Biosystems). Reactions were run in 384-well plate under uniform cycling conditions. The cDNA templates were mixed with the reaction mix and equal volumes (10 μl) was loaded to each well. Cycling conditions were as follows: Uracil N-glycosylase (UNG) activation (50˚C, 2 min), polymerase activation (95˚C for 20 sec), 40 PCR cycles (denaturation, 95˚C for 1 sec and an annealing step performed at 60˚C for 20 sec). Data analysis was based on the ΔΔCT method with normalization of the raw data to a housekeeping gene (actin). Relative quantification is used to compare the gene expression levels between different groups of mice and results were expressed as fold-changes (assuming control group average expression as unit).

## In vitro lipase activity assays

Conditioned media containing recombinant human EL and LPL were generated using recombinant adenoviruses as previously described [14]. Subconfluent COS cells, were infected with recombinant adenoviruses in serum-free medium (3,000 particles/cell). After 48 hours, heparin was added to a final concentration of 10 U/ml to release lipases from the cells surface. The plates were incubated for an additional 30 minutes. The media were then harvested, aliquoted and frozen at −80˚C. Expression of enzymes was confirmed by Western blotting.

For *in vitro* TG and PC lipase activity assays using dual labeled TRL-like emulsions, activity was measured according to a modification of the method of Nilsson-Ehle and Schotz that we have adapted previously [48, 54]. Conditioned medium containing LPL and EL was used as the enzyme source whereas a glycerol-stabilized large lipid emulsion was used as substrate. Briefly, 300 mg triolein (Sigma Aldrich), $^3$H- Triolein (Perkin Elmer 99 μg per 300 mg

nonradioactive triolein), 17.16 mg egg phosphatidylcholine (Sigma Aldrich), and $^{14}$C-dipalmi-toyl-phosphatidylcholine ($^{14}$C-DPPC; 23.38 μmol $^{14}$C-DPPC, giving 1.0 μCi/μmol total DPPC specific activity) were combined in a glass vial and dried under nitrogen. 5 mL of glycerol (Fisher Scientific) were added and the mixture was sonicated using a Branson 450 microtip sonicator for five minutes. Concentrated emulsions were allowed to clear overnight. 15 μL of this concentrated emulsion was combined with 90 μL distilled water, 15 μl of 1.0 M Tris pH 8.0, 15 μl of 15% BSA solution, and 15 μl of 3.0 M NaCl to give a final volume of 150 μl of working emulsions.

Working emulsion substrate (150 μl) was combined with different dilutions of EL and LPL conditioned media (150 μl) or post-heparin plasma (20 μl, diluted 1:8) were incubated at 37˚C for 30 minutes. Purified apoC-II (Fisher Scientific) was added to provide apoC-II for LPL reaction (0.375μg/reaction). The reaction was terminated by the addition of 3.25 ml of methanol: chloroform:heptane solvent (1.41:1.25:1.00). 1.05 ml of pH 10.0 Buffer (Thermo Fisher Scientific) was added and tubes were spun at 2,000 RPM for 20 minutes. The upper phase containing liberated fatty acids was used for scintillation counting (0.5 ml per sample) [55]. The relative amount of hydrolysis of $^{3}$H-Triolein to $^{3}$H-oleate and $^{14}$C-DPPC to $^{14}$C-palmitate was calculated for each sample.

For lipase assays of $^{3}$H-triolein labeled human VLDLs with recombinant EL and LPL, $^{3}$H-triolein labeled VLDL was generated as described above for the intravenous $^{3}$H-triolein TRL clearance in vivo experiments. Purified $^{3}$H-triolein labeled VLDL after dialysis were incubated with recombinant WT EL, recombinant LPL, or both enzymes at equal molar combinations for 4 hours at 37˚C. Reactions were stopped with 100% ethanol, samples were incubated at -20˚C for 2 hours and then ultracentrifuged at 13,500 RPM for 20 minutes to precipitate proteins and separate lipid containing supernatants. The supernatants were run on thin layer chromatography using hexane:diethyl ether:acetic acid buffer (170:30:1) to separate lipid species and fatty acids containing bands were cut and measured for $^{3}$H activity by scintillation counting.

## Statistical analysis

All discrete data from human samples shown are represented as mean ± standard error of the mean (S.E.M.), where error bars show S.E.M. All individual data points are also shown wherever feasible. All time course comparisons across experimental groups for Figs 3–7, S1 and S2 Figs were performed using repeated factor 2-way analysis of variance (2-way ANOVA) unless otherwise described in the figure legends. Statistical significance was defined as P < 0.05 for all analyses. Data graphs in Figs 1 and 2 were generated using R software version 3.5 (https://www.r-project.org/). Data graphs for all other figures were generated using the GraphPad Prism software.

## Supporting information

**S1 Fig. VLDL-TG secretion in *Lipg*$^{-/-}$ mice. A.** Plasma triglycerides in mice at the indicated timepoints following intraperitoneal injection of poloxamer P407, a competitive inhibitor of LPL activity. Plasma TGs were measured by colorimetric biochemical assays. **B.** VLDL-TG secretion rates calculated from the slope of the curves from (A) using the 1, 2, and 4 hour timepoints. **P<0.01, repeated factor 2-way ANOVA comparing WT and *Lipg*$^{-/-}$ groups (A). *P<0.05, student's unpaired T-test comparing WT and *Lipg*$^{-/-}$ groups (B). Data is expressed as mean ± S.E.M.
(TIF)

**S2 Fig. Reconstitution of EL in the liver of *Lipg*<sup>-/-</sup> mice. A-F.** Plasma cholesterol (A), HDL-C (B), nonHDL-C (C), triglycerides (D), phospholipids (E), and nonesterified fatty acids (F) in WT mice treated with AAV Null, *Lipg*<sup>-/-</sup> mice treated with AAV Null, and *Lipg*<sup>-/-</sup> mice treated with AAV murine *Lipg* for 5 weeks after 4 weeks of feeding a high fat diet. Plasma was collected after 4 hours of fasting and lipids were measured by autoanalyzer. $^*P<0.05$, $^{**}P<0.01$, $^{***}P<0.001$, repeated measure 2-way ANOVA compared to the WT AAV Null group. Data is expressed as mean ± S.E.M.
(TIF)

**S1 Table. Lipidomic measurement of TG species concentrations during OFTT.** Concentrations of plasma TG species were measured by LC-MS as described in the Materials and Methods. TG species were measured from plasma of WT vs *Lipg*<sup>-/-</sup> mice after overnight fasting and 7 hours after OFTT with olive oil gavage (10 μl/g). Each column under a group header (WT or *Lipg*<sup>-/-</sup>) corresponds to an individual mouse. Fold Difference corresponds to the ratio of mean concentrations for a given TG species from *Lipg*<sup>-/-</sup> vs WT mice for a given timepoint. P-value corresponds to comparison of WT vs *Lipg*<sup>-/-</sup> mice for a given timepoint by student's unpaired T-test.
(XLSX)

**S2 Table. Lipidomic measurement of PC species concentrations during OFTT.** Concentrations of plasma PC species were measured by LC-MS as described in the Materials and Methods. PC species were measured from plasma of WT vs *Lipg*<sup>-/-</sup> mice after overnight fasting and 7 hours after OFTT with olive oil gavage. Each column under a group header (WT or *Lipg*<sup>-/-</sup>) corresponds to an individual mouse. Fold Difference corresponds to the ratio of mean concentrations for a given PC species from *Lipg*<sup>-/-</sup> vs WT mice for a given timepoint. P-value corresponds to comparison of WT vs *Lipg*<sup>-/-</sup> mice for a given timepoint by student's unpaired T-test.
(XLSX)

**S1 Data. Raw data for Figs 1–7 and S1 and S2 Figs.** Units and measures correspond to the indicated figure panels in the Main Text, Figures or Supporting Information.
(XLSX)

## Acknowledgments

The authors thank the participants of the UKBB, GLGC, and MVP studies for their participation in the respective studies.

## Author Contributions

**Conceptualization:** Sumeet A. Khetarpal, Cecilia Vitali, Daniel J. Rader.

**Data curation:** Sumeet A. Khetarpal, Cecilia Vitali.

**Formal analysis:** Sumeet A. Khetarpal, Cecilia Vitali, Michael G. Levin, Derek Klarin, Joseph Park, Akhil Pampana, Pradeep Natarajan.

**Funding acquisition:** Sumeet A. Khetarpal, Cecilia Vitali, Daniel J. Rader.

**Investigation:** Sumeet A. Khetarpal, Cecilia Vitali, Michael G. Levin, Derek Klarin, Joseph Park, Akhil Pampana, John S. Millar, Takashi Kuwano, Dhavamani Sugasini, Papasani V. Subbaiah, Jeffrey T. Billheimer, Pradeep Natarajan.

**Methodology:** Sumeet A. Khetarpal, Cecilia Vitali, John S. Millar, Papasani V. Subbaiah.

**Resources:** Pradeep Natarajan.

**Supervision:** Sumeet A. Khetarpal, Jeffrey T. Billheimer, Pradeep Natarajan, Daniel J. Rader.

**Validation:** Sumeet A. Khetarpal.

**Visualization:** Sumeet A. Khetarpal.

**Writing – original draft:** Sumeet A. Khetarpal, Daniel J. Rader.

**Writing – review & editing:** Sumeet A. Khetarpal, Cecilia Vitali, Daniel J. Rader.

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
