## [Decision Letter · Decision Letter 0]

2 Jun 2021

Dear Dr Rader,

Thank you very much for submitting your Research Article entitled 'Endothelial lipase mediates efficient lipolysis of triglyceride-rich lipoproteins' to PLOS Genetics.

The manuscript was fully evaluated at the editorial level and by independent peer reviewers. The reviewers appreciated the attention to an important topic and were generally positive. They also identified some concerns, mainly pertaining to data interpretation and statistical analysis, that we ask you address in a revised manuscript.

We therefore ask you to modify the manuscript according to the review recommendations. Your revisions should address the specific points made by each reviewer.

[LINK]

Yours sincerely,

Xia Yang

Associate Editor

PLOS Genetics

Gregory Barsh

Editor-in-Chief

PLOS Genetics

Reviewer's Responses to Questions

**Comments to the Authors:**

Reviewer #1: Khetarpal et al. have performed a detailed study of the role of endothelial lipase (EL) in the catabolism and clearance of triglyceride-rich lipoproteins (TRL). This is a nice study, with strong supporting human genetic data from meta-analyses of data from large consortia. The in vivo EL loss-of-function data showing increased plasma TG and reduced clearance of TRL from the circulation are convincing. However, the data supporting the hypothesis that EL and LPL collaborate synergistically to enhance catabolism of TRL are somewhat weak. My specific comments are detailed below.

1. The association of LIPG p.Asn396Ser with plasma cholesterol, HDL-C, and LDL-C is extremely significant (p=10e-74, p=10e-199, p=10e-65), however the association with TG is somewhat weaker, although still genome-wide significant (p=10e-10). Is this because VLDL is the lipoprotein with the lowest concentration of phospholipid?

2. Why was there only a very modest increase in phospholipids compared to TG (Fig. 3A)? Why were the TG not affected by loss of EL in the fasted animals, unlike HDL-C and LDL-C?

3. Can the authors address why it took 4 weeks of high fat feeding for the TG levels to diverge, when other lipid parameters were affected at baseline (Fig. 3D)? Why were the changes in LDL-C so dramatic in high fat-fed animals (Fig. 3F)?

4. Fig. 5B, left hand graph has groups labeled WT and Lipg–/–, whereas the right hand graph has them labeled WT and EL KO – I assume EL KO refers to Lipg–/–. These should be relabeled for consistency.

5. The enhancement in TG lipase activity (Fig. 6H), whilst statistically significant, is modest. As such, the data supporting the author’s hypothesis “…that EL may collaborate with LPL…” is somewhat weak.

6. The authors use the catalytic mutant p.Ser169Ala in the studies in Fig. 6. What about the effects of the human p.Asn396Ser variant? These studies would support the role of this variant in regulating phospholipase activity.

7. As the authors state that “…EL is expressed throughout the vascular endothelium where LPL is located…” and “…that EL may collaborate with LPL…”, are there any humans with LOF variants in both LPL and LIPG?

8. Loss=of-function studies in mice lacking both LPL and EL would strengthen the study.

Reviewer #2: This is an important study demonstrating for the first time the role of EL in TLR catabolism. The authors provide compelling human and mouse data implicating EL in plasma TG metabolism. The manuscript is well written, the experiments are well designed and an impressive amount of high quality results is clearly and rigorously presented.

A major contention of the study is that EL facilitates TG lipolysis in concert with LPL and the two enzymes act synergistically. However, this conclusion is not supported by the data provided. EL has TG lipase activity (Fig. 6E) and it clearly acts additively with LPL activity (Fig. 6H). Thus, there appears to be no synergism between the two enzymes and the model of EL’s phospholipase activity promoting TG lipolysis by LPL is not born out by the in vitro data. Furthermore, the accumulation of PUFA-containing TGs in EL deficiency (Fig. 6B) suggests that it is EL’s TG lipase activity that is responsible for elevated plasma TG in vivo. Thus, contrary to the model shown in Fig. 7, this study seems to demonstrate that EL contributes to TRL metabolism through its TG lipase activity. The manuscript should be revised accordingly.

Minor comments:

Fig. S1: According to the figure legend, one-way ANOVA was used to analyze data in S1A. This is likely to be a typo, as repeated measure two-way ANOVA should have been used. Also, the calculation of slopes in S1B is unclear. According to Methods, the 0, 60 and 240 min time-points were used. However, the curve is clearly not linear at across this time span. It would be more appropriate to use the 60, 120 and 240 time points for slope estimation.

Fig. 3, Page 8: The authors claim that the mouse HF data suggest “delayed clearance of remnant TRLs in the absence of EL function”. It is unclear what points to reduced remnant clearance. These data seem to indicate reduced TRL clearance.

Fig. S2: Fasting TG levels before AAV injection are almost double in EL-KO vs WT mice in this expriment. This is in sharp contrast to data in Fig. 3A, where no difference is seen. The apparently contradicting results should be explained.

Fig. S2: The meaning of double diamonds is unclear. The figure legend states that these indicate T-test significance between AAV treatments. This is clearly not the case and should be clarified. All time course data, including WT vs EL-KO should be analyzed the same way, preferably by repeated measure two-factor ANOVA.

Fig. 4 and 5: Again, it is unclear how one-way ANOVA, as stated in the figure legend was used in 4A and 4C. Two-way ANOVA should be used to analyze all time course data including those in 4E. The same comment applies to Fig. 5A and 5C. In 5E, LPL data are clearly not normally distributed. Non-parametric analysis should be used instead of T-test.

Fig. 5, Page 9: Again, it is unclear why the authors “hypothesized the delay in remnant TRL clearance”, when it is TG hydrolysis that is affected in EL deficiency.

Fig. 7: Most FA released by LPL (and likely EL) action is taken up by the tissues where hydrolysis takes place (i.e. muscle, adipose). Showing FFA exclusively taken up by liver is misleading.

Discussion: pLOF should be defined.

Reviewer #3: Dear Authors,

I congratulate you all on a nice study blending human genetics with biochemical characterization of endothelial lipase. One aspect of the manuscript that could use clarification is uptake of fatty acids by the liver. In its current written format it comes across that there is impairment of fatty acid uptake. I believe what you mean is that EL allows for cleavage of particle followed by uptake of oleate in the liver. This could be amended in the results and discussion to read clear.

**Have all data underlying the figures and results presented in the manuscript been provided?**

Reviewer #1: None

Reviewer #2: None

Reviewer #3: Yes

PLOS authors have the option to publish the peer review history of their article (what does this mean?). If published, this will include your full peer review and any attached files.

Reviewer #1: No

Reviewer #2: No

Reviewer #3: No

---

## [Decision Letter · Decision Letter 1]

10 Aug 2021

Dear Dr. Rader,

Many thanks for making a thorough revision of your manuscript! The reviewers are in general satisfied with the revised manuscript. However, Reviewer 2 has raised a number of suggestions that will help further improve your manuscript. Please address as many of Reviewer 2's points as possible and submit a minor revision within 30 days.

[LINK]

Yours sincerely,

Xia Yang

Associate Editor

PLOS Genetics

Gregory Barsh

Editor-in-Chief

PLOS Genetics

Reviewer's Responses to Questions

**Comments to the Authors:**

Reviewer #1: Khetarpal et al. have responded to the prior review with detailed and thoughtful responses. They have performed additional experiments and should be commended for their thorough consideration of all the reviewers' comments.

This is now a very nice, complete article, with important insights for the field. Well done!

Reviewer #2: The authors addressed the reviewers’ questions in detail and substantially revised and improved the manuscript. Importantly, they included new data that demonstrate the interaction between EL and LPL in TRL hydrolysis in vitro and in vivo. Addressing the following issues would further strengthen the manuscript:

1. Line 245: Reference to Fig. 6G is missing.

2. Line 246: The reference to 30 min is confusing, as EL increases TG-lipase activity up to 100 min.

3. Lines 248-250: This language suggests that Ser169Ala somehow affects LPL. This is clearly not the case, as it also reduces TG lipase activity in the absence of LPL (Fig. 6E). It should be stated that DN EL likely inhibits endogenous EL produced by COS7 cells.

4. How do the authors explain that DN EL inhibits the TG-lipase activity of endogenously produced EL (Figs. 6E and 6H), but fails to reduce (6F) or even increases (6G) PL activity? These results should be explained, or at least acknowledged.

5. Fig. 6F: Figure shows PL activity, but the figure legend states TG-lipase.

6. Fig. 6I is missing from the figure legends.

7. Fig. S3: These data provide critical in vivo support for one of the main contentions of the manuscript, i.e. the collaboration between EL and LPL. As such, these figures would be more appropriate in the main body of the paper.

8. Figs. 6I and S3F: The use of one-sample T-test is inappropriate, because it assumes a known theoretical value as a comparison and ignores the variation in measured values. Thus, this test will inflate the significance of differences. In Fig. 6I, one-way ANOVA should be used to evaluate the difference of EL+LPL from the other two groups. The large increase above the sum of means (red line) provides strong support for an interaction between the two lipases, even without further statistical analysis. In Fig. S3F, the significance of interaction term (genotype x AAV) in 2-way ANOVA should be assessed.

9. Fig. S3E: The TG scale seems to be off by a factor of 10.

Reviewer #3: .

**Have all data underlying the figures and results presented in the manuscript been provided?**

Reviewer #1: None

Reviewer #2: None

Reviewer #3: Yes

PLOS authors have the option to publish the peer review history of their article (what does this mean?). If published, this will include your full peer review and any attached files.

---

## [Decision Letter · Decision Letter 2]

2 Sep 2021

Dear Dr Rader,

We are pleased to inform you that your manuscript entitled "Endothelial lipase mediates efficient lipolysis of triglyceride-rich lipoproteins" has been editorially accepted for publication in PLOS Genetics. Congratulations!

Yours sincerely,

Xia Yang

Associate Editor

PLOS Genetics

Gregory Barsh

Editor-in-Chief

PLOS Genetics

Comments from the reviewers (if applicable):

Reviewer's Responses to Questions

**Comments to the Authors:**

Reviewer #2: All outstanding questions have been addressed in the revised manuscript. Congratulations to an excellent study.

**Have all data underlying the figures and results presented in the manuscript been provided?**

Reviewer #2: None

PLOS authors have the option to publish the peer review history of their article (what does this mean?). If published, this will include your full peer review and any attached files.

Reviewer #2: No

**Data Deposition**

http://datadryad.org/submit?journalID=pgenetics&manu=PGENETICS-D-21-00599R2

**Press Queries**

---

## [Editor Report · Acceptance letter]

16 Sep 2021

PGENETICS-D-21-00599R2 

Endothelial lipase mediates efficient lipolysis of triglyceride-rich lipoproteins 

Dear Dr Rader, 

We are pleased to inform you that your manuscript entitled "Endothelial lipase mediates efficient lipolysis of triglyceride-rich lipoproteins" has been formally accepted for publication in PLOS Genetics! Your manuscript is now with our production department and you will be notified of the publication date in due course.

With kind regards,

Katalin Szabo

PLOS Genetics

On behalf of:
